# RL in Latent MDPs is Tractable:
# Online Guarantees via Off-Policy Evaluation

**Jeongyeol Kwon**
University of Wisconsin-Madison
jeongyeol.kwon@wisc.edu

**Shie Mannor**
Technion / NVIDIA AI
shie@ee.technion.ac.il

**Constantine Caramanis**
University of Texas at Austin
constantine@utexas.edu

**Yonathan Efroni**
Meta AI
jonathan.efroni@gmail.com

## Abstract

In many real-world decision problems there is partially observed, hidden or latent information that remains fixed throughout an interaction. Such decision problems can be modeled as Latent Markov Decision Processes (LMDPs), where a latent variable is selected at the beginning of an interaction and is not disclosed to the agent. In the last decade, there has been significant progress in solving LMDPs under different structural assumptions. However, for general LMDPs, even in the tabular case, no algorithm is known to provably match the existing lower bound [41]. We introduce the first sample-efficient algorithm for LMDPs without *any additional distributional assumptions*. Our result builds off a new perspective on the role of off-policy evaluation guarantees and coverage coefficients in LMDPs, a perspective, that has been overlooked in the context of exploration in partially observed environments. Specifically, we establish a novel off-policy evaluation lemma and introduce a new coverage coefficient for LMDPs. Then, we show how these can be used to derive near-optimal guarantees of an optimistic exploration algorithm. These results, we believe, can be valuable for a wide range of interactive learning problems beyond LMDPs, and especially, for partially observed environments.

## 1 Introduction

In Reinforcement Learning (RL) [54], an agent aims to maximize the long-term cumulative rewards through interactions within an *unknown* environment. Markov Decision Processes (MDPs) are perhaps the most well-studied and popular framework for this goal. As the name suggests, MDPs heavily rely on the Markovian assumption that requires the state to be fully observable. However, many real-world decision problems involve critical partially observed or latent information, such as sensitive or unknown preference information of users in recommendation systems [28], undiagnosed illness in medical treatments [62, 53], and adaptation to uninformed tasks in robotics [67, 48]. Even when such latent factors remain fixed throughout a period of interactions the fundamental Markovian property of MDPs is no longer valid.

A line of work has proposed efficient RL algorithms in the presence of latent contexts [12, 24, 11, 21, 41, 37] within the framework that we here collectively refer to as Latent Markov Decision Processes (LMDP) following [41]. In LMDPs, nature selects an MDP from a finite set of $M$ candidate MDP models at the beginning of a period of interactions (a.k.a. episode), and an agent interacts with the chosen MDP for $H$ time steps of an episode (the horizon). However, the identity of the chosen MDP is not given to the agent. We call this unknown identity the *latent context*.

38th Conference on Neural Information Processing Systems (NeurIPS 2024).

Most prior work on LMDPs has relied on strict separation assumptions (*e.g.,* [11, 24, 37]). The applicability of these approaches is limited to scenarios where the horizon is sufficiently large and identification of the latent model can be guaranteed, *i.e.,* $H \gg \Omega(SA)$ [24], where $S$ and $A$ are the state and action spaces size. Without these explicit horizon requirements, as far we know, all existing algorithms suffer *the curse of horizon*, requiring sample complexity $\Omega(A^H)$ – which frequently arises in the more general framework of Partially Observed MDPs (POMDPs) [52, 39]. Without the ability to identify the underlying latent model, it remains unclear how to address the curse of horizon inherent in partially observed systems [39].

Recently, a series of works [40, 42, 43] proposed sample efficient algorithms without separation assumptions when $M = O(1)$, assuming the transition dynamics of models with different latent context is similar. While this is still a substantial contribution, their results cannot be easily extended to the general LMDP setting with different transition dynamics (see Section 1.1). Consequently, to date, the following question has remained open:

*Can we break the curse of horizon in LMDPs if $M = O(1)$ without any assumptions?*

In this work we provide the first sample-efficient exploration algorithm for LMDPs without any assumptions. Throughout the paper, we assume that $H > 2M$, and focus on whether we can improve the trivial upper bound that incurs complexity $\Omega(A^H)$. Since a $\Omega(SA)^M$ lower bound for LMDPs has been established [41], our goal is to achieve an upper bound of $\mathrm{poly}(S, A)^M$ without any assumptions, namely, to get a matching upper bound up to polynomial factors.

## 1.1 Technical Challenges

Many online RL algorithms follow a similar pattern. They make use of a confidence set – a set of candidate models (hypothesis) that can explain the observed data with high probability – and execute a policy that will shrink the volume of the confidence set is produced and executed [6, 38, 27, 7, 33, 36]. The entirety of the statistical problem is to analyze the decaying rate of confidence sets under proper model class assumptions [49, 34, 19].

**Challenge 1: Limitation of Existing POMDP Algorithms.** Existing approaches for online exploration in partially observed systems largely fall into the category of Optimistic Maximum Likelihood Estimation (OMLE) [45, 46]. This class of algorithms often requires an assumption that allows the construction of shrinking confidence sets. These algorithms also assume access to a set of special policies – called *core-tests* – to be executed to generate trajectories [5, 8, 16, 17, 45, 57, 13, 22, 46, 26]. Without specifying the proper core-tests, the volume of confidence sets may not decay in a desired rate, leading to the curse of horizon $\Omega(A^H)$ [39, 15]. Further, existing POMDP approaches require an ability to recover the belief of the underlying model from observations, *e.g.,* by assuming the distribution of observations when executing the core-tests is invertible to the belief over hidden states. Consequently, existing literature on POMDPs has two limitations: (i) it requires to specify *a priori* a set of core-tests policies, and (ii) it assumes the full-rankness of the state-observation emission matrix when the core-tests are being executed.

While LMDPs are a special class of POMDPs, neither the existence of a set of core-tests is known *a priori*, nor it is possible to recover the belief over latent contexts from distribution of trajectories (see Section B for details). This creates a fundamental challenge for existing approaches when applied to LMDPs. Further, little is understood on learning a near-optimal policy among "doubly-exponential" number of candidate history-dependent policies without either the visibility of contexts or core-tests. This calls for a new perspective on the question of efficient exploration in LMDPs.

**Challenge 2: Limitation of Existing LMDP Algorithms.** The work of [43] suggested an alternative strategy to learn a near-optimal policy in LMDPs: the moment-matching approach for exploration in LMDPs. When all contexts share the same state-transition dynamics, the notion of moments can be defined as the joint distribution of rewards under *a fixed prior* at a tuple of at most $d := 2M - 1$ state-action pairs $\boldsymbol{x} = \big( (s_{[1]}, a_{[1]}), ..., (s_{[d]}, a_{[d]}) \big)$. This in turn suggests that the exploration algorithm must learn how to visit these length-$d$ state-action tuples simultaneously, *i.e.,* find a policy that ensures that $\boldsymbol{x}$ appears as a subsequence of the entire trajectory with high enough probability.

When the transition dynamics of different latent contexts is similar, reaching optimally to $d$ state-action pairs is a minor challenge; *e.g.,* we can first learn the shared transition kernel with any reward-free

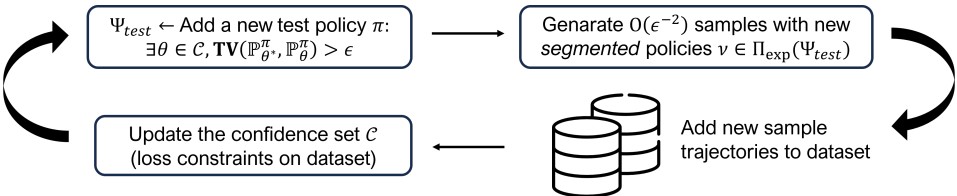

**Figure 1:** Highlevel description of `LMDP-OMLE`. In the online phase, we find a new test policy under which models in the confidence set do not agree. Then the exploration policy is constructed with our new notion of *segmentation* of policies within $\Psi_{\texttt{test}}$ that are executed throughout. In the offline phase, we add the batched sample trajectories to dataset and update the confidence set of models.

exploration scheme for MDPs [33], and then execute the policy that maximizes the probability of reaching the $d$ state-action pairs. However, for general LMDP, when the transition dynamics of different latent contexts may differ, this approach is no longer available since the latent transition dynamics may not be learnable in general. Furthermore, to follow the notion of moments suggested in [43], the data collected for estimating the correlation tensor must be collected under the same prior (belief) over all latent contexts. Unfortunately, ensuring this for general LMDPs, when the transition dynamics of different latent contexts are not equal, is impossible, since even if we obtain the samples of correlations, different policies may result in different and unknown priors over contexts. These challenges hint we need an alternative approach to solve general LMDPs, when the transition dynamics vary between latent contexts.

**Challenge 3: Limitations of Existing Complexity Measures in RL.**   Numerous studies have examined complexity measures for RL with function approximation or in the rich-observation settings [30, 34, 19]. These studies are based on the Markovian assumption, which does not hold in the LMDP setting where the entire history may be needed to decode the latent state. When defining the effective state as the entire history at each time step, it is unclear how to analyze the complexity measures from these studies without resorting to exponential guarantees in the horizon.

## 1.2   Overview of Our Contribution

Recent studies have found some fundamental connections between off-policy evaluation (OPE) and online exploration in RL [59, 2, 29, 9, 55, 3, 4]. In this work, we offer a fresh viewpoint, which deviates from existing works, on the connection between OPE and online exploration. This perspective, together with new analysis tools, allows us to provide a sample-efficient algorithm for the LMDP setting. This further showcases the usefulness of OPE for online exploration in POMDPs.

Arguably, the fundamental question in OPE is the following: how much does a behavioral policy $\psi$ tell about a target policy $\pi$? The simplest form of the OPE guarantee in MDPs relies on the notion of *coverage coefficient* given by:

$$C(\psi;\pi) = \max_{s,a,t} \frac{\mathbb{P}^\pi(s_t = s, a_t = a)}{\mathbb{P}^\psi(s_t = s, a_t = a)}.$$

How would this quantity be related to online exploration? A key observation to start developing intuition is the following: an unbounded coverage coefficient, *i.e.,* $C(\psi;\pi) = \infty$ implies there exists a state-action pair, at some time-steps, that cannot be reached under $\psi$, but can be reached with $\pi$.

The algorithmic framework we develop in this work builds off `OMLE` [46]. In Section 3, we consider the MDP setting to provide intuition of our analysis. There, `OMLE` iteratively tests new policies on models from the confidence set which predict different outcomes, until the trajectory distribution of all policies is reliably estimated. Since the number of new state-action pairs is bounded for MDPs, the number of times the coverage coefficient can be large must be bounded during an interaction. We provide new analysis for the MDP setting based on OPE tools.

To apply this approach for LMDPs, we are required to develop a new notion of coverage coefficient and new OPE tools. We propose a coverage coefficient that can be informally described as follows:

$$C(\psi;\pi) = \max_{(\mathcal{E},\mathcal{I})} \max_m \frac{\mathbb{P}^\pi(\mathcal{T} \in \mathcal{E} \mid m)}{\mathbb{P}^\psi(\mathcal{T} \in \mathcal{E} \mid m, \mathbf{do}\ \mathcal{I})}$$

where $m$ is the unobserved latent context, $\mathcal{T} = (s_t, a_t, r_t)_{t \in [H]}$ is a sampled trajectory, $\mathcal{E}$ is an event of interest, *e.g.,* visiting length of at most $d$ tuples of states and actions within an episode, and $\mathcal{I}$ is an intervention of interest, *e.g.,* force an action $a$ at the $t^{th}$ time step regardless of $\psi$ (for the formal definition, see Definition 4.1). Note that the coverage coefficient cannot be measured explicitly, since $m$ is a latent variable; nevertheless, this concept is central to our analysis and our ability to analyze the sample complexity of the proposed algorithm. Its usefulness lies in an OPE guarantee we develop (see Lemma 4.2):

$$\texttt{TV}(\mathbb{P}_{\theta^*}^\pi, \mathbb{P}_\theta^\pi)(\mathcal{T}) \lesssim C(\psi; \pi) \cdot \sum_{\mathcal{I}} \texttt{TV}(\mathbb{P}_{\theta^*}^\psi, \mathbb{P}_\theta^\psi)(\mathcal{T} \mid \mathbf{do}\ \mathcal{I}),$$

where $\texttt{TV}(\mathbb{P}_1, \mathbb{P}_2)(\cdot)$ is the total-variation (TV) distance between two probability measures $\mathbb{P}_1, \mathbb{P}_2$.

With these tools at hand, we design an iterative online exploration algorithm for the LMDP setting, and prove its sample complexity matches the lower bound, up to polynomial factors. The algorithm, we refer as `LMDP-OMLE` (see Figure 1 for highlevel illustration), repeats the following: *(i)* find a policy for which the trajectory distributions between models in the confidence set is large or terminate, or *(ii)* collect new data with exploration policies constructed with a set of (obtained) test policies and interventions, an exploration strategy for LMDPs that we introduce.

## 2 Preliminaries

We consider an episodic RL with time-horizon $H$ in LMDPs defined as follows:

**Definition 2.1 (Latent Markov Decision Process (LMDP))** *An LMDP $\mathcal{M}$ consists of a tuple $(\mathcal{S}, \mathcal{A}, \mathcal{R}, \theta, H)$ with a state space $\mathcal{S}$; action space $\mathcal{A}$; reward space $\mathcal{R}$, and a finite-time horizon $H$. $\theta$ is a model parameter consisting of multiple MDPs in the model $\theta := (\{w_m, T_m, R_m\})_{m=1}^M$. In each $m^{th}$ MDP, $T_m : \mathcal{S} \times \mathcal{A} \times \mathcal{S} \to [0, 1]$ maps a state-action pair and a next state to a probability; $R_m : \mathcal{S} \times \mathcal{A} \times \mathcal{R} \to [0, 1]$ is a probability of rewards; $\{w_m\}_{m=1}^M$ are the mixing weights such that at the beginning of every episode the $m^{th}$ model is chosen with probability $w_m$.*

Without loss of generality, we assume that there exists a null state that represents the starting and terminal state $s_0 = s_{H+1} = \emptyset$, and a null action at the beginning of an episode $a_0 = \emptyset$, even though actual policies do not take any action at the beginning. $T_m(\cdot|s_0, a_0)$ is the initial state distribution of the $m^{th}$ MDP. We assume that the number of latent contexts is constant $M = O(1)$, and the time-horizon is larger than the number of contexts $H > 2M$. Further, we assume the reward values are finite and bounded:

**Assumption 2.2 (Finite and Bounded Reward)** *The reward distribution has finite support with (arbitrarily large) cardinality, and each reward is bounded: $|r| \leq 1$ for all $r \in \mathcal{R}$.*

We also note that this concept can be easily generalized to instantaneous observations that include rewards, and thus, we do not lose much generality due to Assumption 2.2. We consider a policy class $\Pi$ which contains all history-dependent policies $\pi : \Xi \times (\mathcal{S}, \mathcal{A}, \mathcal{R})^* \times (\mathcal{S} \times [H]) \to \Delta(\mathcal{A})$, where $\Xi$ is the space of independent variables decided at the beginning of execution. As a special case, we consider the class of memoryless policies: $\Pi_{\text{mls}} : (\mathcal{S} \times [H]) \to \Delta(\mathcal{A})$ We are interested in finding an optimal history dependent policy $\pi \in \Pi$ that maximizes the expected reward: $V_{\theta^*}^* := \max_{\pi \in \Pi} \mathbb{E}_{\theta^*}^\pi \left[ \sum_{t=1}^H r_t \right]$, where $\theta^* \in \Theta$ is the true model parameter and $\mathbb{E}_{\theta^*}^\pi[\cdot]$ is expectation taken over the true LMDP model $\mathcal{M}^*$ when policy $\pi$ is executed.

**Notation** We use $[n] := \{1, \ldots, n\}$ and $[n]_+ := \{0\} \cup [n]$. We define $d := 2M - 1$ and assume $H > 2M$. Let $\texttt{SubSeq}(H, d)$ be the set of subsequences of $(1, 2, ..., H)$ with length less than or equal to $d$, *i.e.,* $\texttt{SubSeq}(H, d) := \{(\tau_1, \tau_2, ..., \tau_q) | q \in [d], 1 \leq \tau_1 < ... < \tau_q \leq H\}$. We often denote a state-action pair $(s, a)$ as one symbol $x = (s, a) \in \mathcal{X} = (\mathcal{S} \times \mathcal{A})$, and an reward-next state pair $(r, s')$ as one symbol $y = (r, s') \in \mathcal{Y} = (\mathcal{R} \times \mathcal{S})$. We often express the next state at time step $t$ as either $s_{t+1}$ or $s'_t$, and the pair of instantaneous observation and next state as $y_t = (r_t, s_{t+1}) = (r_t, s'_t)$. For any segment of a sequence $(z_1, z_2, ..., z_H)$ from $t_1$ to $t_2$, we often simplify the notation as $z_{t_1:t_2}$. We denote the entire trajectory as $\mathcal{T} := (s, a, r)_{1:H}$, and $\mathcal{T}_{1:t} = ((s, a, r)_{1:t-1}, s_t)$ for a history of length $t$. For any set $\mathcal{S}$, we define $\mathcal{S}^{\otimes k}$ as a short-hand for the $k$-times Cartesian power of $\mathcal{S}$.

---

**Algorithm 1** `MDP-OMLE`

---

1: **Input:** $n_{\text{test}} \in \mathbb{N}, \beta, \epsilon_{\text{TV}}, \eta > 0, \mathcal{C}^0 = \Theta$
2: Initialize $k = 0$
3: **while** there exists $\pi^k \in \Pi_{\text{mls}}$, and $\theta_1, \theta_2 \in \mathcal{C}^k$ such that $\text{TV}\left(\mathbb{P}_{\theta_1}^{\pi^k}, \mathbb{P}_{\theta_2}^{\pi^k}\right)(\mathcal{T}) > 4\epsilon_{\text{TV}}$ **do**
4:     Generate data $\{\mathcal{T}_j^k\}_{j=1}^{n_{\text{test}}}$ by executing $\pi^k$, update $\mathcal{D}^k \leftarrow \mathcal{D}^{k-1} \cup \{(\mathcal{T}_j^k, \pi^k)\}_{j=1}^{n_{\text{test}}}$
5:     Refine the confidence set with the dataset:

$$\mathcal{C}^{k+1} = \left\{\theta \in \Theta \Big| \sum_{(\mathcal{T},\pi) \in \mathcal{D}^k} \log \mathbb{P}_\theta^\pi(\mathcal{T}) \geq \arg\max_{\theta \in \Theta} \sum_{(\mathcal{T},\pi) \in \mathcal{D}^k} \log \mathbb{P}_\theta^\pi(\mathcal{T}) - \beta\right\} \quad (1)$$

    $k \leftarrow k + 1$
6: **end while**
7: Pick any $\theta \in \mathcal{C}^k$ and return the optimal policy of $\mathcal{M} := (\mathcal{S}, \mathcal{A}, \mathcal{O}, \theta)$.

---

We define $\texttt{SubTraj}(\mathcal{T}, \boldsymbol{\tau}) \subseteq (\mathcal{X} \times \mathcal{Y})^{\otimes |\boldsymbol{\tau}|}$ as a valid subsequence of trajectories at time-steps $\boldsymbol{\tau} \in \texttt{SubSeq}(H, d)$, *i.e.*, if $(x_{\boldsymbol{\tau}}, y_{\boldsymbol{\tau}}) \in \texttt{SubTraj}(\mathcal{T}, \boldsymbol{\tau})$, for any $i$ such that $\tau_i = \tau_{i+1}, y_{\tau_i} = (r_{\tau_i}, s'_{\tau_i})$ and $x_{\tau_{i+1}} = (s_{\tau_{i+1}}, a_{\tau_{i+1}})$ must have $s'_{\tau_i} = s_{\tau_{i+1}}$.

For a tuple of state-action pairs (or states) of length $q$, we denote $\boldsymbol{x} = (x_{[1]}, ..., x_{[q]})$ (or $\boldsymbol{s} = (s_{[1]}, ..., s_{[q]})$ with bracketed indices for each element to distinguish from time steps. We use $|\boldsymbol{x}|$ for the length of sequence $\boldsymbol{x}$. We denote the cardinality of the state and action space as $S := |\mathcal{S}|$ and $A := |\mathcal{A}|$. For any two models $\theta_1, \theta_2$, we often denote $\mathbb{P}_1(\cdot) := \mathbb{P}_{\theta_1}(\cdot)$ and $\mathbb{P}_2(\cdot) := \mathbb{P}_{\theta_2}(\cdot)$ whenever the context is clear. We denote $P_m(\cdot)$ for a probability measured conditioned on the context $m \in [M]$ over the ground-truth model ($\theta_1$ when we compare $\theta_1$ and $\theta_2$). We denote $\texttt{Unif}(\mathcal{A})$ as the uniform distribution over a set $\mathcal{A}$. Let $\texttt{TV}(\mathbb{P}_1, \mathbb{P}_2)(X)$ be the total-variation distance between two probability measures $\mathbb{P}_1(\cdot), \mathbb{P}_2(\cdot)$ over a random variable $X$.

## 3   New Perspective on `OMLE`: Online Guarantees via Off-Policy Evaluation

In this section, we present our new approach for analyzing the `OMLE` algorithm, and, for establishing intuition in the Markovian setting. Differently than prior analysis [45, 46] which is based on the generalized eluder-type condition assumption (see [46], Condition 3.2), we show that a certain type of an OPE guarantee can be used to study the performance of `OMLE`. This alternative perspective is instrumental in designing a sample-efficient algorithm for the LMDP class.

Consider `MDP-OMLE` depicted in Algorithm 1. `MDP-OMLE` is an adaptation of `OMLE` for the MDP setting with the goal of learning a near-optimal policy. The algorithm iteratively refines the confidence set, i.e., the set of statistically valid models, until it terminates. Specifically, it iteratively repeats the two steps: *(i)* find a policy for which the TV distance between trajectory distributions of models in the confidence set is sufficiently large, and *(ii)* collect data with that policy, and use the data to refine the confidence set. To bound the sample complexity of the algorithm we attempt to upper bound the number of iterations, namely, to bound the number of times the TV distance between trajectory distributions can be sufficiently large.

The following OPE lemma is a tool that allows us to bound the number of iterations of `MDP-OMLE`. Before discussing its application, we present the result.

**Lemma 3.1 (TV Bound via OPE for MDPs)** *For any behavioral and target policies $\psi, \pi \in \Pi$, let the coverage coefficient be defined by:*

$$C(\psi; \pi) = \max_{t \in [H]} \max_{x \in \mathcal{X}} \frac{\mathbb{P}_{\theta^*}^\pi(x_t = x)}{\mathbb{P}_{\theta^*}^\psi(x_t = x)}. \quad (2)$$

*For any two models $\theta, \theta^* \in \Theta$, the TV distance between trajectory distributions following a target policy $\pi \in \Pi$ is bounded as follows:*

$$TV(\mathbb{P}_{\theta^*}^\pi, \mathbb{P}_\theta^\pi)(\mathcal{T}) \leq 2C(\psi; \pi) \sum_{t \in [H]} TV(\mathbb{P}_{\theta^*}^\psi, \mathbb{P}_\theta^\psi)(x_t, y_t). \quad (3)$$

How can we use this result to bound the number of iterations of `MDP-OMLE`? Consider the infinite sample regime, when `MDP-OMLE` collects infinite data at each iteration by executing a policy $\pi^k$ on the

$k$th iteration, *i.e.*, $n_{\texttt{test}} = \infty$. Further, assume the algorithm is at the beginning of its $k+1$ iteration. In the infinite sample regime all models in the confidence set must have matching event distribution relatively to the underlying model measured when policy $\pi^k$ is tested. Specifically, for all $\theta \in \mathcal{C}^k$ and $t \in [H]$ it holds that $\text{TV}(\mathbb{P}_{\theta^*}^{\pi^k}, \mathbb{P}_{\theta}^{\pi^k})(x_t, y_t) = 0$. Then Lemma 3.1 implies the following: for all policies $\pi$ for which $C(\pi^k; \pi) < \infty$ it also holds that $\text{TV}(\mathbb{P}_{\theta^*}^{\pi}, \mathbb{P}_{\theta}^{\pi})(\mathcal{T}) = 0$. Conversely, assume the condition of the while loop at the beginning of the $k+1$ iteration holds true, namely, there exists a policy $\bar{\pi}$ for which $\text{TV}(\mathbb{P}_{\theta^*}^{\bar{\pi}}, \mathbb{P}_{\theta}^{\bar{\pi}})(\mathcal{T}) > 0$. Then Lemma 3.1 also implies that $C(\pi^k; \bar{\pi}) = \infty$, namely, there exists an $x \in \mathcal{X}$ and $t \in [H]$ such that $\mathbb{P}_{\theta^*}^{\bar{\pi}}(x_t = x) > 0$ whereas $\mathbb{P}_{\theta^*}^{\pi^k}(x_t = x) = 0$. Next, recall that MDP-OMLE sets the data collection policy at the $k+1$ iteration to be $\pi^{k+1} = \bar{\pi}$. Hence, the data collection policy at the $k+1$ iteration will visit some state-action pair at some time step $\pi^k$ did not visit. Hence, in the infinite sample regime, MDP-OMLE halts after at most $HSA$ iterations, as there are at most $HSA$ different state-action pairs in different time steps.

The intuition presented above is robust to sampling error, *i.e,* when $n_{\texttt{test}} < \infty$. To simplify the discussion, let us temporarily assume that $\mathbb{P}_{\theta^*}^{\pi}(x_t = x) > \gamma$ for all $\pi \in \Pi$ and $x \in \mathcal{S} \times \mathcal{A}$ (we do not require this assumption in our final result by analyzing a perturbed MDP). The key intuition on which the finite sample analysis builds upon is formalized in the following lemma:

**Lemma 3.2 (Coverage Multiplicative Increase)** *For all $k > 0$ in Algorithm 1, there exists at least one $t \in [H]$ and $x \in \mathcal{X}$ such that*

$$\mathbb{P}_{\theta^*}^{\pi^k}(x_t = x) \geq c \cdot \frac{\epsilon_{TV}}{H} \sqrt{\frac{n_{test}}{(HSA)\beta}} \cdot \max_{j < k} \mathbb{P}_{\theta^*}^{\pi^j}(x_t = x).$$

*with some absolute constant $c > 0$.*

Therefore, by setting the number of samples to be $n_{\texttt{test}} \geq (4H^2 SA\beta)/(c\epsilon_{\texttt{TV}})^2$, we ensure that in every iteration MDP-OMLE doubles the coverage of at least one state-action pair at a certain time step. Therefore, the algorithm terminates within at most $K = O(HSA \cdot \log(1/\gamma))$ iterations with high probability . After termination, we are guaranteed that any two models in the confidence set are $\epsilon_{\texttt{TV}}$-close in TV-distance for any policy, hence we can obtain $\epsilon = (H\epsilon_{\texttt{TV}})$-optimal policy. To summarize, we state the following theorem:

**Theorem 3.3** *Let $K = O(HSA) \log(HSA/\epsilon)$ and $\beta = \log(K|\Theta|/\eta)$. Then, with probability at least, $1 - \delta$, MDP-OMLE terminates after $K$ iterations with at most $N$ episodes being generated, where*

$$N \geq O(H^6 S^2 A^2) \cdot \log(HSA/\epsilon) \log(K|\Theta|/\eta)/\epsilon^2,$$

*and outputs an $\epsilon$-optimal policy with probability at least $1 - \eta$.*

In a typical tabular MDP setting, we take $O(\log |\Theta|) = \tilde{O}(SA)$, by discretizing the class of MDPs. Hence the sample complexity of MDP-OMLE is $N = \tilde{O}(H^6 S^3 A^3/\epsilon^2)$. While this upper bound is suboptimal compared to the minimax rate [7], the appeal of this type of analysis is its ability to bypass the need for analyzing the decaying volume of the constructed confidence sets (Section 1.1, Challenge 1).

## 4 Efficient Exploration in LMDPs

In previous section we presented a new approach to analyze the OMLE algorithm for MDPs. Next, we develop an analogous technique for the LMDP setting and design the LMDP-OMLE algorithm. Central to its design and analysis is an OPE lemma and a new coverage coefficient which we now present.

**Intuition from moment-exploration algorithm in [43].** Before we dive into our key results, let us provide our intuition on how we construct the OPE lemma for LMDPs. Our construction is inspired by the moment-exploration algorithm proposed in [43]: when state-transition dynamics are identical across latent contexts, *i.e.*, $T_1 = T_2 = ... = T_M$, we can first learn the transition dynamics with any reward-free type exploration scheme for MDPs [33], and then set the exploration policy that sufficiently visits some tuples of state-actions $\boldsymbol{x}$ of length at most $d$. Specifically, they set a memoryless exploration policy $\psi \in \Pi_{\texttt{mls}}$ which sets $\mathbb{P}^{\psi}(x_{\boldsymbol{\tau}} = \boldsymbol{x})$ sufficiently large for some $\boldsymbol{\tau} \in \texttt{SubSeq}(H, d)$

---

**Algorithm 2** `LMDP-OMLE`

---

1: **Input:** $K, d, n_{\text{test}} \in \mathbb{N}$, $\epsilon_{\text{test}}, \eta > 0$, $\beta = \log(K|\Theta|/\eta)$, $\Psi_{\text{test}}^0 = \{\text{Unif}(\mathcal{A})\}$, $\mathcal{D}^0 = \{(\mathcal{T}_j, \text{Unif}(\mathcal{A}))\}_{j=1}^{n_{\text{test}}}$ where each $\mathcal{T}_j$ is generated by executing $\text{Unif}(\mathcal{A})$, $\mathcal{C}^1$ as defined in (1)
2: Initialize $k = 1$
3: **while** there exists $\pi^k \in \Pi_{\text{mls}}$, and $\theta_1, \theta_2 \in \mathcal{C}^k$ such that $\text{TV}\left(\mathbb{P}_{\theta_1}^{\pi^k}, \mathbb{P}_{\theta_2}^{\pi^k}\right)(\mathcal{T}) > 4\epsilon_{\text{test}}$ **do**
4:     $\Psi_{\text{test}}^k \leftarrow \Psi_{\text{test}}^{k-1} \cup \{\pi^k\}$, initialize $\mathcal{D}^k = \mathcal{D}^{k-1}$
5:     **for** all $\boldsymbol{\psi} = (\psi_0, \psi_1, ..., \psi_d) \in \Psi_{\text{test}}^k \times ... \times \Psi_{\text{test}}^k \times \{\text{Unif}(\mathcal{A})\}$ with at least one $i \in [d-1]_+$ being $\psi_i = \pi^k$, and $\boldsymbol{\tau} \in \text{SubSeq}(H, d)$, $\boldsymbol{z} \in \{0,1\}^{|\boldsymbol{\tau}|}$ **do**
6:         Generate data $\{\mathcal{T}_j\}_{j=1}^{n_{\text{test}}}$ by executing $\nu(\boldsymbol{\psi}; \boldsymbol{\tau}, \boldsymbol{z})$
7:         Update $\mathcal{D}^k \leftarrow \mathcal{D}^k \cup \{(\mathcal{T}_j, \nu(\boldsymbol{\psi}; \boldsymbol{\tau}, \boldsymbol{z}))\}_{j=1}^{n_{\text{test}}}$
8:     **end for**
9:     Update the confidence set $\mathcal{C}^{k+1}$ according to equation (1)
10:    $k \leftarrow k + 1$
11: **end while**
12: Pick any $\theta \in \mathcal{C}^k$ and return the optimal policy of $\mathcal{M} := (\mathcal{S}, \mathcal{A}, \mathcal{R}, \theta)$.

---

and $\boldsymbol{x} \in \mathcal{X}^{\otimes |\boldsymbol{\tau}|}$. We note that the same moment-exploration strategy cannot be applied to general LMDPs with different state-transition dynamics since learning the transition dynamics itself involves latent contexts. Nevertheless, the intuition from [43] suggests that our key statistics are this visitation probabilities to all tuples of state-actions within a trajectory.

## 4.1 Off-Policy Evaluation in LMDPs

The OPE lemma we derive in this section makes use of a behavior policy of a special form which we refer to as a *segmented policy*, inspired by the notion of moment-exploration in [43]. Let us formally define the key quantities to establish our OPE lemma. A segmented policy, which we denote by $\nu(\boldsymbol{\psi}; \boldsymbol{\tau}, \boldsymbol{z})$, takes as an input a sequence of history-dependent policies, $\boldsymbol{\psi} = (\psi_0, ..., \psi_d)$, a sequence of time steps, we call checkpoints, $\boldsymbol{\tau} = (\tau_1, ..., \tau_{|\boldsymbol{\tau}|}) \in \text{SubSeq}(H, d)$, and a sequence of binary numbers $\boldsymbol{z} = (z_1, ..., z_{|\boldsymbol{\tau}|}) \in \{0,1\}^{|\boldsymbol{\tau}|}$ where $|\boldsymbol{\tau}| \leq d$, and returns a history-dependent policy.

The segmented policy $\nu(\boldsymbol{\psi}; \boldsymbol{\tau}, \boldsymbol{z})$ switches sequentially between different policies in $\boldsymbol{\psi}$. The time steps in which the switch occurs are determined by $\boldsymbol{\tau}$: starting from time step $\tau_i + 1$ policy $\psi_i$ will be executed. Finally, the sequence $\boldsymbol{z}$ determines whether an intervention with a random action will occur at the $\tau_i$ time-step. If $z_i = 1$ the executed action at time step $\tau_i$ is the uniform action, $\text{Unif}(\mathcal{A})$, and, otherwise, the policy $\psi_{i-1}$ is executed. The segmented policy is also denoted by

$$\nu(\boldsymbol{\psi}; \boldsymbol{\tau}, \boldsymbol{z}) := \psi_0 \underset{(\tau_1, z_1)}{\circ} \psi_1 \underset{(\tau_2, z_2)}{\circ} ... \underset{(\tau_{|\boldsymbol{\tau}|}, z_{|\boldsymbol{\tau}|})}{\circ} \psi_{|\boldsymbol{\tau}|},$$

where "$\pi_a \underset{(t,z)}{\circ} \pi_b$" means switch to policy $\pi_b$ at starting from time step $t + 1$, and at time step $t$ take random action if $z = 1$ and otherwise execute $\pi_a$.

We are now ready to define a coverage coefficient for the LMDP class of models. This new coverage coefficient is central to the analysis and design of `LMDP-OMLE`.

**Definition 4.1 (LMDP Coverage Coefficient)** *The LMDP coverage coefficient of a sequence of policies $\boldsymbol{\psi} \in \Pi^{\otimes(d+1)}$ with respect to a target policy $\pi \in \Pi$ in is given by:*

$$C(\boldsymbol{\psi}; \pi) := \max_{\boldsymbol{\tau} \in SubSeq(H,d)} \max_{\boldsymbol{z} \in \{0,1\}^{\otimes |\boldsymbol{\tau}|}} \max_{(\boldsymbol{x}, \boldsymbol{y}) \in SubTraj(\mathcal{T}, \boldsymbol{\tau})} \max_{m \in [M]} \frac{P_m^\pi(x_{\boldsymbol{\tau}} = \boldsymbol{x}, y_{\boldsymbol{\tau}} = \boldsymbol{y})}{P_m^{\nu(\boldsymbol{\psi}; \boldsymbol{\tau}, \boldsymbol{z})}(x_{\boldsymbol{\tau}} = \boldsymbol{x}, y_{\boldsymbol{\tau}} = \boldsymbol{y})}. \quad (4)$$

The LMDP coverage coefficient $C(\boldsymbol{\psi}; \pi)$ between a sequence of policies, $\boldsymbol{\psi}$, and a target, history-dependent, policy $\pi$, depends on the worst-case way to generate a segmented policy, $\nu(\boldsymbol{\psi}; \boldsymbol{\tau}, \boldsymbol{z})$ from $\boldsymbol{\psi}$. Further, it is a worst-case ratio of the probability of a sequence of observations within $|\boldsymbol{\tau}| = d$ different time steps, namely, $x_{\boldsymbol{\tau}}, y_{\boldsymbol{\tau}}$. This is different than the standard coverage coefficient (see equation (2)), that depends on observation from a single time. Fortunately, $C(\boldsymbol{\psi}; \pi)$ requires only a partial set of observations, instead of using full trajectories. This is crucial towards developing

sample complexity guarantees that are not exponential in $H$. Lastly, observe that the LMDP coverage coefficient depends on the latent context $m$, and thus, we cannot measure $C(\psi; \pi)$ from samples.

We are now ready to provide the key OPE lemma, which makes use of the LMDP coverage coefficient.

**Lemma 4.2 (TV Bound via OPE for LMDPs)** *Let $d = 2M - 1$. For any two models $\theta, \theta^* \in \Theta$, and for any $\pi \in \Pi$ and $\psi \in \Pi^{\otimes(d+1)}$, let $C(\psi; \pi)$ be defined as (4) over $\theta^*$. Then the following holds:*

$$TV(\mathbb{P}_{\theta^*}^\pi, \mathbb{P}_\theta^\pi)(\mathcal{T}) \leq M \cdot C(\psi; \pi) \sum_{\tau \in SubSeq(H,d)} \sum_{z \in \{0,1\}^{\otimes |\tau|}} TV\left(\mathbb{P}_{\theta^*}^{\nu(\psi;\tau,z)}, \mathbb{P}_\theta^{\nu(\psi;\tau,z)}\right)(x_\tau, y_\tau).$$

This result is analgous to the OPE result for MDPs (see Lemma 3.1). It is a tool that allows us to bound the TV distance between trajectory distributions of a history-dependent policy $\pi$ by a term that depends on a segmented policy $\nu(\psi; \tau, z)$ and an LMDP coverage coefficient. Importantly, the term on the RHS that depends on the segmented policy, $\nu(\psi; \tau, z)$, is a sum of distributions of partial trajectories of size $|\tau| \leq d$, which is independent of the horizon length, $H$.

**Remark 4.3 (Why is single latent-state coverability coefficient not enough?)** *One may wonder why it is not sufficient to consider a single latent-state coverability analogous to Lemma 3.1, namely an analogous to (2) defined as:*

$$\max_{t \in [H]} \max_{x \in \mathcal{X}} \max_{m \in [M]} \frac{P_m^\pi(x_t = x)}{P_m^\psi(x_t = x)}.$$

*In Appendix D.5 we provide a counter-example where such single latent-state coverage coefficient is finite, and yet, off-policy evaluation guarantee cannot be established.*

## 4.2 Coverage Doubling via Sufficiency of Memoryless Polices

To convert the OPE guarantee to an online exploration algorithm, we aim to use the coverage-doubling argument. Ideally, we could apply the coverage-doubling argument with the general policy class similarly to the MDP case as presented in Section 3. However, in its current form, Lemma 4.2 requires the behavior policy to be a segmented policy, and is not valid for any general behavioral policy. Hence, it is not obvious on which probabilistic events we can apply the coverage doubling argument. We leave it as future work whether we can obtain an off-policy evaluation lemma with general history-dependent behavioral policies, and its clearer conversion to online guarantees.

In this work, we present an alternative plan to the above issue: we reduce the search space from history-dependent policies to memoryless policies. This allows us to track quantities on a *segmentwise* coverage. Specifically, we first note that the LMDP coverage coefficient can be bounded (after maximizing over the sequence $z$) by:

$$C(\psi; \pi) \leq \max_{\tau \in SubSeq(H,d)} \max_{\substack{x \in \mathcal{X}^{\otimes |\tau|} \\ s' \in \mathcal{S}^{\otimes |\tau|-1}}} \max_{m \in [M]} \prod_{i=0}^{d-1} \frac{\max_{\mathcal{T}_{1:\tau_i}} P_m^\pi(s_{\tau_{i+1}} = s_{[i+1]} | s'_{\tau_i} = s'_{[i]}, \mathcal{T}_{1:\tau_i})}{(1/A) \cdot P_m^{\nu(\psi_i;\tau_i)}(s_{\tau_{i+1}} = s_{[i+1]} | s'_{\tau_i} = s'_{[i]})}, \quad (5)$$

where $\nu(\psi_i; \tau_i)$ denotes a segmented policy executing $\psi_i$ after the $\tau_i^{th}$ time-step with memory reset, hence ignoring the history up to $\tau_i$ (the conditioning event $s'_{\tau_0} = s'_{[0]}$ at $i = 0$ can be ignored). Inspired by the form in denominator, we aim to double the following probability defined over a *context-segment* pair:

$$\max_{\psi \in \Psi_{\text{test}}} P_m^{\nu(\psi;t_1)}(s_{t_2} = s | s'_{t_1} = s'), \quad (6)$$

for at least one $m \in [M], s, s' \in \mathcal{S}$ and $t_1 < t_2$. However, another challenge remains: the RHS in equation (5) consists of the maximum over all possible histories in the numerator, whereas in the denominator we force the data collection policy to reset the memory at checkpoints. We still have to side-step this discrepancy to apply the coverage doubling argument.

The restriction to the class of memoryless policies allows us to resolve these issues since

$$\max_{\mathcal{T}_{1:t_1}} P_m^\pi(s_{t_2} = s | s'_{t_1} = s', \mathcal{T}_{1:t_1}) = P_m^\pi(s_{t_2} = s | s'_{t_1} = s'),$$

$$\text{and } P_m^{\nu(\psi;t_1)}(s_{t_2} = s | s'_{t_1} = s') = P_m^{\psi}(s_{t_2} = s | s'_{t_1} = s'),$$

if $\pi, \psi \in \Pi_{\text{mls}}$ since $P_m$ represents the latent Markovian transition dynamics. Thus, we can aim to double up the quantity in equation (6). To apply this argument, we establish our second key lemma, a crucial building block for the coverage doubling argument:

**Lemma 4.4 (Sufficiency of Memoryless Polices for LMDPs)** *Suppose the following holds:*

$$\max_{\pi_{\text{mls}} \in \Pi_{\text{mls}}} TV(\mathbb{P}_{\theta^*}^{\pi_{\text{mls}}}, \mathbb{P}_{\theta}^{\pi_{\text{mls}}})(\mathcal{T}) \leq \epsilon_{test}. \tag{7}$$

*Then for any history-dependent policies $\pi \in \Pi$, the following holds:*

$$TV(\mathbb{P}_1^{\pi}, \mathbb{P}_2^{\pi})(\mathcal{T}) \leq M(2H^2)^d \cdot (MSA)^d \cdot \epsilon_{test}.$$

Therefore, we reduced our goal of learning an optimal policy to finding a set of models which satisfies equation (7) with respect to all memoryless policies. Importantly, upon estimating the trajectory distribution up to accuracy $\epsilon_{test} > 0$ for memoryless policies, we have a bounded TV distance between the trajectory distribution of all history-dependent policies, includes the optimal policy.

### 4.3 The `LMDP-OMLE` Algorithm

Once the search space is reduced to memoryless policies, we aim to match trajectory distributions for all memoryless policies. At a high-level, `LMDP-OMLE` follows similar recipe to `MDP-OMLE`, and is summarized in Algorithm 2. It can be described as follows:

1. Find a memoryless policy $\pi^k \in \Pi_{\text{mls}}$ whose prediction on trajectory distributions does not match between two models in the confidence set $\mathcal{C}^k$. Add $\pi^k$ to the collection of test policies $\Psi_{\text{test}}^k$, that forms each segment of (segmented) exploration policies.

2. Collect new sample trajectories following the new set of segmented policies for exploration, generated by different combinations of collected test policies and switching operations.

3. Update the confidence set $\mathcal{C}^k$ with Maximum Likelihood Estimation (MLE) on the updated dataset $\mathcal{D}^k$ by equation (1).

The data collection policy of `LMDP-OMLE` (second step above) is inspired and leverages Lemma 4.2 to give upper bounds on the TV distance of untested policies. Next, by Lemma 4.4, we know that when the while loop terminates, any optimal policy of a model contained in the confidence set is a near-optimal policy of the underlying LMDP. We conclude this section with our main theorem on the sample complexity of learning the optimal policy in Latent MDPs:

**Theorem 4.5 (Sample Complexity of `LMDP-OMLE`)** *Let $d = 2M - 1$ and assume $H > 2M$. After at most $K = O(MS^2H) \cdot \log(MSAH/\epsilon)$ iterations, `LMDP-OMLE` (Algorithm 2) terminates with at most $N$ episodes being generated where*

$$N \gtrsim \left(M^4 S^6 A^4 H^7 \cdot \log(MSAH/\epsilon)\right)^d \cdot M^4 H^2 \cdot \log(K|\Theta|/\eta)/\epsilon^2,$$

*and outputs an $\epsilon$-optimal policy with probability at least $1 - \eta$.*

Note that in tabular LMDPs with finite support rewards, we have $\log(|\Theta|) = O(S^2A|\mathcal{R}|\log(1/\epsilon))$. The appeal of $\log(|\Theta|)$ dependence is an flexible extension of the same result to parameterized reward distributions. In Section D.1 and D.4 we provide a proof overview and a full proof of Theorem 4.5.

## 5 Conclusion and Future Work

In this work, we presented the first sample-efficient algorithm for LMDPs, resolving an open question of efficient exploration with latent contexts. While our result is specialized to LMDPs, we believe our new perspectives and techniques on deriving online guarantees through the lens of OPE can be useful for a broader range of interactive learning, and, especially, partially observed problems. While resolving the open problem, there are a few remaining questions for the LMDP setting.

**Tightness of the Result.** The upper bound in Theorem 4.5 scales with $\tilde{O}\left(MSAH \cdot \log(1/\epsilon)\right)^{O(M)}$, while the existing lower bound is $\Omega(SA)^M$. Closing this polynomial gap in the exponent, and having a matching upper and lower bounds can be valuable for a deeper understanding of LMDPs and possibly for POMDPs in general.

**General OPE lemma and Regret Guarantees for LMDPs.** The OPE lemma derived in this work (Lemma 4.2) assumes the behavior policy is a segmented policy with intervention at different checkpoints. While this result allows us to provide guarantees on `LMDP-OMLE` and prove it learns a near-optimal policy, this result is restrictive, in that it does not provide general guarantees for OPE nor makes it possible to derive regret guarantees. In particular, can we evaluate $\pi \in \Pi$ without policy-switching or intervention when the behavioral policy is a generic history-dependent policy $\psi \in \Pi$? Further, is there an algorithm with provable $\text{poly}(S, A)^M \cdot \sqrt{T}$ regret for the general LMDP setting?

**Towards Practical Settings.** Our result gives a worst-case guarantee. Yet, practical instances may be much simpler under different set of assumptions *e.g.,* with provided side-information [66, 44] or additional structural assumptions [41, 63, 14]. Deriving new conditions can be of great importance for real-world applications, *e.g.,* there could be more practical notion of separation, or the set of instances that allows the notion of coverage-coefficient with a (significantly) shorter length $d = o(M)$ of state-action tuples. Further, developing practical RL methodologies for the LMDP setting remains an unexplored challenge with significant importance for numerous applications. These are remained to be explored in future works.

### Acknowledgment

This research was partially funded by AFOSR/AFRL grant no. FA9550-18-1-0166, NSF Grants 2019844 and 2112471, and Israel Science Foundation Grant No. 2199/20.

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

# Appendix A    Additional Preliminaries

The difference between the values of any policy $\pi \in \Pi$ measured on two models $\theta_1, \theta_2 \in \Theta$ are bounded by the total-variation (TV) distance between trajectory distributions, that is,

$$|V_1^\pi - V_2^\pi| \leq H \cdot \text{TV}(\mathbb{P}_1^\pi, \mathbb{P}_2^\pi)(\mathcal{T}),$$

since the maximum reward that can be obtained in an episode is bounded by $H$ due to Assumption 2.2. Hence, if we can show that

$$\text{TV}(\mathbb{P}_{\theta^*}^\pi, \mathbb{P}_\theta^\pi)(\mathcal{T}) \leq \epsilon/H =: \epsilon_{\text{TV}}, \qquad \forall \pi \in \Pi, \tag{8}$$

then an optimal policy $\hat{\pi}^*$ of the empirical model $\hat{\theta}$ is guaranteed to be $2\epsilon$-optimal in the true model $\theta^*$. Henceforth, we focus on finding an empirical model $\hat{\theta}$ that satisfies (8).

To bound the TV-distance between trajectory distributions for all history-dependent policies $\pi \in \Pi$ between any two LMDP models, we start by unfolding the expression of statistical distance

$$\text{TV}(\mathbb{P}_1^\pi, \mathbb{P}_2^\pi)(\mathcal{T}) = \sum_{(x_t, r_t)_{t \in [H]}} |\mathbb{P}_1^\pi((x_t, r_t)_{t \in [H]}) - \mathbb{P}_2^\pi((x_t, r_t)_{t \in [H]})|$$

$$= \sum_{\substack{x_{1:H} \\ r_{1:H}}} \prod_{t=1}^H \pi(a_t | \mathcal{T}_{1:t}) \times \left| \sum_{m=1}^M w_m^1 \prod_{t=0}^H T_m^1(s_{t+1}|x_t) R_m^1(r_t|x_t) - \sum_{m=1}^M w_m^2 \prod_{t=0}^H T_m^2(s_{t+1}|x_t) R_m^2(r_t|x_t) \right|.$$

When the context is clear, we compare trajectory distributions between any two given model parameters $\theta_1, \theta_2 \in \Theta$, and denote the probability measure from each model following $\pi$ as $\mathbb{P}_1^\pi(\cdot), \mathbb{P}_2^\pi(\cdot)$.

## A.1    Additional Notation

To reduce the notation overload, we use $P_m(y_t|x_t) = T_m(s_{t+1}|x_t) R_m(r_t|x_t)$. When the context is clear, we often use a shorthand $\pi_t = \pi(a_t | \mathcal{T}_{1:t})$, and denote $\pi_{t_1:t_2}$ as a shorthand of the product of a time-consecutive sequence from $t_1$ to $t_2$, i.e., $\pi_{t_1:t_2} = \prod_{t=t_1}^{t_2} \pi_t$. When we sum over both $x_t$ and $y_t$, we implicitly mean that the $s_t'$ part of $y_t$, which we denote as $s'(y_t)$, must match to the $s_{t+1}$ part of $x_{t+1}$, which we denote as $s(x_{t+1})$. Using the notation, we rewrite the unfolded TV-distance equation as the following:

$$\sum_{x_{1:H}} \sum_{y_{1:H}} \pi_{1:H} \left| \sum_{m=1}^M w_m^1 \prod_{t=0}^H P_m^1(y_t|x_t) - \sum_{m=1}^M w_m^2 \prod_{t=0}^H P_m^2(y_t|x_t) \right|.$$

We use a shorthand $\delta_\pi(X)$ for $|\mathbb{P}_1^\pi(X) - \mathbb{P}_2^\pi(X)| = |\sum_{m=1}^M w_m^1 P_m^{1,\pi}(X) - \sum_{m=1}^M w_m^2 P_m^{2,\pi}(X)|$, and thus $\sum_X \delta_\pi(X) = \text{TV}(\mathbb{P}_1^\pi, \mathbb{P}_2^\pi)(X)$ where the summation is over all possible realizations of a random variable $X$. Finally, we denote $d = 2M - 1$.

## A.2    Preliminaries for Lemma 4.2

Here we define a few more quantities that will be crucial for the proofs for Section 4. In bounding the total variation distance in terms of tested policies without exponential blow-up in $H$, the key is to marginalize events across time steps. Let us first fix the *checkpoint* time-steps $\boldsymbol{\tau} = (\tau_1, ..., \tau_q)$ for $q \in [d]$, and a sequence of executable policies $\boldsymbol{\psi} = (\psi_0, \psi_1, ..., \psi_d)$. Each $i^{th}$ segment policy will be executed within time interval $(\tau_i, \tau_{i+1}]$ for $i \geq 0$.

To proceed, for the initial policy $\psi_0$, let $l^0$ be the smallest quantity, among the contexts, of the ratio between the state visitation probabilities in consecutive steps when $\psi_0$ is executed (recall that we denote $x_t = (s_t, a_t)$, $y_t = (r_t, s_{t+1})$):

$$l^0(x_t, r_t; s_{t+1}) := \min_{n \in \{1,2\}} \left( \min_{m \in [M_n]} \frac{P_m^{n, \psi_0}(s_t)}{P_m^{n, \psi_0}(s_{t+1})} P_m^n(r_t, s_{t+1}|x_t) \right). \tag{9}$$

This quantity can be understood as the minimum (over latent contexts) of the "pseudo-posterior" probabilities of 1-step event given the future state if $\psi_0$ is memoryless, since

$$l^0(x_t, r_t; s_{t+1}) \cdot \psi_0(a_t|s_t) = \min_{n \in \{1,2\}} \min_{m \in [M]} P_m^{\psi_0}(x_t, r_t|s_{t+1}), \text{ if } \psi_0 \in \Pi_{\text{mls}}.$$

Henceforth we use a shorthand $l_t^0 := l^0(x_t, r_t; s_{t+1})$. We recursively define a sequence of the above quantity.

Next, we fix the event $(x_{\boldsymbol{\tau}}, y_{\boldsymbol{\tau}})$ at the event-log time-steps. For all $i \geq 0$ and $\tau_i < t \leq \tau_{i+1}$, we define $l_t^i(x_{\tau_{1:i}}, y_{\tau_{1:i}})$ and $p_m^{n,i}(x_{\tau_{1:i}}, y_{\tau_{1:i}})$ recursively as the following:

$$l_t^i(x_{\tau_{1:i}}, y_{\tau_{1:i}}) := \min_{n \in \{1,2\}} \left( \min_{m \in [M]: p_m^{n,i}(x_{\tau_{1:i}}, y_{\tau_{1:i}}) > 0} \frac{P_m^{n, \nu(\psi_i; \tau_i)}(s_t | s_{\tau_i+1}) P_m^n(r_t, s_{t+1} | x_t)}{P_m^{n, \nu(\psi_i; \tau_i)}(s_{t+1} | s_{\tau_i+1})} \right), \quad (10)$$

and

$$p_m^{n,i+1}(x_{\tau_{1:i+1}}, y_{\tau_{1:i+1}}) = p_m^{n,i}(x_{\tau_{1:i}}, y_{\tau_{1:i}}) \times \left( P_m^{n, \nu(\psi_i; \tau_i)}(s_{\tau_{i+1}} | s_{\tau_i+1}) P_m^n(y_{\tau_{i+1}} | x_{\tau_{i+1}}) \right.$$
$$\left. - P_m^{n, \nu(\psi_i; \tau_i)}(s'(y_{\tau_{i+1}}) | s_{\tau_i+1}) l_{\tau_{i+1}+1}^i(x_{\tau_{1:i}}, y_{\tau_{1:i}}) \right), \quad (11)$$

Here, we define $t_{[0]} \equiv -1$, where we recall that $\nu(\pi; t)$ means the memory reset of a policy $\pi$ at time step $t+1$. $x_{\tau_{1:0}}, y_{\tau_{1:0}} \equiv \phi$, and $l_0^0 \equiv 1$ and $p_m^{n,0} \equiv w_m^n$ for $n = 1, 2$. The key point here is that in this recursive construction, as $i$ increase, we have at least one $i$ such that either $p_m^{1,i+1} = 0$ or $p_m^{2,i+1} = 0$, *i.e.,* at least one context is removed from consideration at each checkpoint.

In the subsequent steps in our proof, we often omit the dependence on $(x_{\tau_{1:i}}, y_{\tau_{1:i}})$, as well as $\pi_{\tau_{0:i}}$ in $l_t^i$ and $p_m^{n,i+1}$ when the context is clear. Finally, we define

$$\Delta(x_{\boldsymbol{\tau}}, y_{\boldsymbol{\tau}}) := \left| \sum_{m=1}^M p_m^{1, |\boldsymbol{\tau}|} - \sum_{m=1}^M p_m^{2, |\boldsymbol{\tau}|} \right|,$$

Now we are ready to state our key intermediate lemma:

**Lemma A.1** *For any target policy $\pi \in \Pi$ and a sequence of segment policies $\boldsymbol{\psi} = (\psi_0, \psi_1, ..., \psi_d)$, the following holds:*

$$\sum_{x_{1:H}} \sum_{y_{1:H}} \pi_{1:H} \left| \sum_{m=1}^M w_m^1 \prod_{t=0}^H P_m^1(y_t | x_t) - \sum_{m=1}^M w_m^2 \prod_{t=0}^H P_m^2(y_t | x_t) \right|$$

$$\leq \sum_{\boldsymbol{\tau} \in SubSeq(H,d)} \sum_{x_{\boldsymbol{\tau}}, y_{\boldsymbol{\tau}}} \Delta(x_{\boldsymbol{\tau}}, y_{\boldsymbol{\tau}}) \times \left( \frac{P_{m(x_{\boldsymbol{\tau}}, y_{\boldsymbol{\tau}})}^{\pi}(x_{\boldsymbol{\tau}}, y_{\boldsymbol{\tau}})}{\prod_{i=0}^{|\boldsymbol{\tau}|-1} P_{m(x_{\boldsymbol{\tau}}, y_{\boldsymbol{\tau}})}^{\nu(\psi_i; \tau_i)}(s_{\tau_{i+1}} | s_{\tau_i+1}) P_{m(x_{\boldsymbol{\tau}}, y_{\boldsymbol{\tau}})}(y_{\tau_{i+1}} | x_{\tau_{i+1}})} \right),$$
$$(12)$$

*where $m(x_{\boldsymbol{\tau}}, y_{\boldsymbol{\tau}})$ is the smallest $m \in [M]$ such that $p_m^{1, |\boldsymbol{\tau}|} > 0$.*

### A.3 Preliminaries for Lemma 4.4

We present additional tools that are useful for proving Lemma 4.4. We first define the notion of (latent) segment coverage coefficient of the *set* of test policies $\Psi_{\text{test}}$ with respect to $\pi$ as the following:

**Definition A.2 (LMDP Segment Coverage Coefficient)** *The coverage of $\pi \in \Pi$ with respect to a set of test policies $\Psi_{test} \subseteq \Pi$ is defined as:*

$$\rho(\Psi_{test}; \pi) := \max_{t_1 < t_2} \max_{s, s'} \max_m \frac{\max_{h:|h|=t_1} P_m^{\pi}(s_{t_2} = s | s'_{t_1} = s', h)}{\max_{\psi \in \Psi_{test}} P_m^{\nu(\psi; t_1)}(s_{t_2} = s | s'_{t_1} = s')}, \quad (13)$$

The key lemma is to bound the coverage coefficient of segmented policies consisting of mixture policies defined as the following:

**Lemma A.3** *Let $\psi_\xi \in \Pi$ be a mixture of a subset of behavioral policies for the following set with a fixed $t_0 \in [H]$:*

$$\Psi_\xi = \left\{ \arg \max_{\psi \in \Psi_{test}} P_m^{\nu(\psi; t_0)}(s_{t_0+t} = s | s'_{t_0} = s'), \forall m, s, s', t \right\} \subseteq \Psi_{test}. \quad (14)$$

*Let $n = |\Psi_\xi|$. We define the mixture policy as $\psi_\xi := \frac{1}{n} \sum_{\psi \in \Psi_\xi} \psi$, i.e., given the time interval $l$ until the next checkpoint time, $\psi_\xi$ first uniformly randomly picks one policy from $\Psi_\xi$ and executes the picked policy afterwards. Let $\boldsymbol{\psi}_\xi := (\psi_\xi, \psi_\xi, ..., \psi_\xi)$ (of length $d + 1$). Then the following holds:*

$$C(\boldsymbol{\psi}_\xi; \pi) \leq (nA \cdot \rho(\Psi_{test}; \pi))^d.$$

## A.4 Auxiliary Concentration Lemmas

The following lemmas are the standard concentration of log-likelihood values of the models within the confidence set. The proofs are standard and can also be in *e.g.,* [1, 45, 44] and [20, 64]. We let $\mathcal{D}^k$ be the dataset at the beginning of the $k^{th}$ iteration in Algorithm 2. We denote $\beta := \log(K|\Theta|/\eta)$.

**Lemma A.4 (Uniform Bound on the Likelihood Ratios)** *With probability $1 - \eta$ for any $\eta > 0$, for all $k \in [K]$ and for any $\theta \in \Theta$,*

$$\sum_{(\mathcal{T},\pi) \in \mathcal{D}^k} \log(\mathbb{P}_\theta^\pi(\mathcal{T})) - \beta \leq \sum_{(\mathcal{T},\pi) \in \mathcal{D}^k} \log(\mathbb{P}_{\theta^*}^\pi(\mathcal{T})). \tag{15}$$

**Lemma A.5 (Concentration of Maximum Likelihood Estimators)** *With probability $1 - \eta$, for all $k \in [K]$, $t \in [H]$ and $\theta \in \Theta$, we have*

$$\sum_{(\mathcal{T},\pi) \in \mathcal{D}^k} TV^2 \left( \mathbb{P}_\theta^\pi, \mathbb{P}_{\theta^*}^\pi \right) (\mathcal{T}) \leq \sum_{(\mathcal{T},\pi) \in \mathcal{D}^k} \log \left( \frac{\mathbb{P}_{\theta^*}^\pi(\mathcal{T})}{\mathbb{P}_\theta^\pi(\mathcal{T})} \right) + 3\beta.$$

# Appendix B    Additional Related Work

The literature on reinforcement learning theory is fast growing. While learning in fully observable systems has been extensively studied in the past decades, relatively little has been understood for online exploration in partially observable systems until recently. We review recent theoretical advances in reinforcement learning with partial observations that are closely related to us.

**Exploration in POMDPs**    Learning a near-optimal policy in POMDPs is a notoriously hard problem [52] due to its full generality. In particular, the statistical complexity of online exploration in a general POMDP fundamentally suffers from the curse of horizon [39]. A earlier breakthrough involved introducing structural assumptions on system dynamics, which enable the recovery of underlying POMDPs model under the uniform ergodicity assumption [25, 5, 8, 23].

Recent theoretical breakthrough concerns the exploration problem in POMDPs without the ergodicity assumption [32, 45], initiating a remarkable progress in understanding the statistical complexity of reinforcement learning in POMDPs under proper structural assumptions. To this date, well-studied tractable POMDP classes (that overcome the curse of horizon) can be considered largely as a system with a "short-window" for efficient exploration [16, 1, 18, 17, 22, 45, 13]. The crux of the short-window assumption is the prior knowledge that the *a short consecutive* execution of purely random actions is enough to obtain sufficient statistics of histories, *i.e.,* the full-rankness of latent state-future observation emission matrices. A short sequence of uniformly random actions become the set of core tests of the system [51]. In such systems, learning the optimal policy only incurs $\text{poly}(S, A) \cdot A^L$ complexity [46] (with window-size $L$), breaking the curse of horizon.

Unfortunately, the same story does not apply in LMDPs, as there is no such "short-window" assumption that allows us to learn the sufficient statistics of histories. This calls for a set of new ideas and concepts, which could be of independent interest.

**Off-Policy Evaluation in POMDPs**    Along with the fast progress in online reinforcement learning with partial observations, there is a growing interest in off-policy evaluation with partial observations [47, 56, 61]. While the sample complexity of OPE has been extensively studied in MDPs under various model class assumptions [31, 60, 58], most existing OPE results in POMDPs are asymptotic in nature, and often suffers the curse of horizon due to their use of importance-weight sampling.

Recently, several recent works have proposed an alternative measure of coverage in the latent space, breaking the curse of horizon [50, 10, 65]. However, their results rely on the weakly-revealing assumptions that is often made in tractable POMDP classes [45], and can only evaluate within the class of memoryless policies. Our results are developed for the off-policy evaluation in LMDPs with several new concepts, which can also be of independent interest to off-policy evaluation problems in partially observed systems.

## B.1 Additional Details on the Full-Rankness Assumption

We give a more detailed explanation of the full-rankness assumption that has become popular in the POMDP literature [45]. As mentioned, the statistical sufficiency of core-tests, which is represented as the minimum singular value of the "latent state-future trajectory" emission matrix $L$. Such an assumption has been exploited in the earlier work of LMDPs in [41], where the matrix $L$ is defined in the following form:

$$L_s[m, (\psi_{\text{test}}, \mathcal{T}_{t:H})] = P_m^{\nu(\psi_{\text{test}};t)}(\mathcal{T}_{t:H}|s),$$

where $\psi_{\text{test}} \in \Psi_{\text{test}}$ is a test policy, and $\mathcal{T}_{t:H}$ is the future trajectory after time step $t$. Therefore, direct application of POMDP approaches such as OMLE require the prior knowledge of $\Psi_{\text{test}}$ and $\sigma_{\min}(L_s) > 0$ for all $s \in \mathcal{S}$. The rationale behind such assumptions is to ensure that a distribution of future trajectories can be converted to a belief of latent contexts, hence a distribution of future observations can serve as an alternative to a belief state. However, we are not given the set of core-tests $\Psi_{\text{test}}$, or even the existence of $\Psi_{\text{test}}$ that ensures $\sigma_{\min}(L_s) > 0$ for general Latent MDPs.

**With Separation.** In a recent work by Chen et al. [14], a polynomial upper bound in $M$ has been established under a notion of strong-separation between contexts with a sufficiently long time horizon $H$. In essence, their assumptions guarantee that $\sigma_{\min}(L_s) > 0$ holds for most of the states with *a priori* given test policy, along with additional analysis for the tail of episodes. It is of great importance to identify such practical assumptions that lead to *fully* polynomial upper bounds, especially for instances with some proper notion of separations even when no prior knowledge of test policies is provided.

# Appendix C  Proofs for Section 3

## C.1  Proof of Lemma 3.1

This base case corresponds to Lemma 4.2 with $M_1 = M_2 = 1$. For convenience, we let $\theta_1 = \theta^*$ and $\theta_2 = \theta$, and thus, $\mathbb{P}_1 = \mathbb{P}_{\theta^*}$ and $\mathbb{P}_2 = \mathbb{P}_\theta$. We can show the inequality recursively:

$$\sum_{x_{1:H}} \sum_{y_{1:H}} \pi_{1:H} \left| \prod_{t=0}^{H} \mathbb{P}_1(y_t|x_t) - \prod_{t=0}^{H} \mathbb{P}_2(y_t|x_t) \right|$$

$$\leq \sum_{x_{1:H}} \sum_{y_{1:H-1}} \pi_{1:H} \left| \prod_{t=0}^{H-1} \mathbb{P}_1(y_t|x_t) - \prod_{t=0}^{H-1} \mathbb{P}_2(y_t|x_t) \right| \sum_{y_H} \mathbb{P}_2(y_H|x_H)$$

$$+ \sum_{x_{1:H}} \sum_{y_{1:H-1}} \pi_{1:H} \prod_{t=0}^{H-1} \mathbb{P}_1(y_t|x_t) \sum_{y_H} |\mathbb{P}_1(y_H|x_H) - \mathbb{P}_2(y_H|x_H)|.$$

Note that $\sum_{y_H} \mathbb{P}_2(y_H|x_H) = 1$ and

$$\sum_{x_{1:H-1}} \sum_{y_{1:H-1}} \pi_{1:H-1} \prod_{t=0}^{H-1} \mathbb{P}_1(y_t|x_t) = \mathbb{P}_1^\pi(s_H),$$

since we implicitly sum over $s_H = s'(y_{H-1})$ as we described in Appendix A.1. Thus,

$$\sum_{x_{1:H}} \sum_{y_{1:H}} \pi_{1:H} \left| \prod_{t=0}^{H} \mathbb{P}_1(y_t|x_t) - \prod_{t=0}^{H} \mathbb{P}_2(y_t|x_t) \right| \leq \sum_{x_{1:H}} \sum_{y_{1:H-1}} \pi_{1:H} \left| \prod_{t=0}^{H-1} \mathbb{P}_1(y_t|x_t) - \prod_{t=0}^{H-1} \mathbb{P}_2(y_t|x_t) \right|$$

$$+ \sum_{x_H, y_H} \mathbb{P}_1^\pi(x_H) |\mathbb{P}_1(y_H|x_H) - \mathbb{P}_2(y_H|x_H)|.$$

Then we can show that

$$\sum_{x_H, y_H} \mathbb{P}_1^\pi(x_H) |\mathbb{P}_1(y_H|x_H) - \mathbb{P}_2(y_H|x_H)| = \sum_{x_H, y_H} \left( \frac{\mathbb{P}_1^\pi(x_H)}{\mathbb{P}_1^\psi(x_H)} \right) \mathbb{P}_1^\psi(x_H) |\mathbb{P}_1(y_H|x_H) - \mathbb{P}_2(y_H|x_H)|$$

$$\leq C(\psi; \pi) \left( \sum_{x_H, y_H} |\mathbb{P}_1^\psi(x_H, y_H) - \mathbb{P}_2^\pi(x_H, y_H)| + \sum_{x_H, y_H} |\mathbb{P}_1^\psi(x_H) - \mathbb{P}_2^\pi(x_H)| \mathbb{P}_2(y_H|x_H) \right)$$

$$\leq 2C(\psi;\pi)\mathrm{TV}(\mathbb{P}_1^\psi,\mathbb{P}_2^\psi)(x_H,y_H).$$

Applying the same step inductively from $t = H$ to $t = 1$, we get the lemma.

## C.2 Proof of Lemma 3.2

Let $\Psi_\xi := \left\{\pi^{j^*(x,t)}, \forall x, t \mid j^*(x,t) := \arg\max_{j\in 0,1,\dots,k-1}\mathbb{P}^{\pi^j}(x_t = x)\right\}$. Then, let $\psi_\xi \in \Pi$ be a policy that can adapt to the predetermined checkpoint $l$, such that $\psi_\xi = \frac{1}{|\Psi_\xi|}\sum_{\psi\in\Psi_\xi}\psi$, *i.e.*, a mixture of policies in $\Psi_\xi$. Note that $|\Psi_\xi| \leq HSA$. Lemma 3.1 tells us that

$$\mathrm{TV}(\mathbb{P}_1^\pi,\mathbb{P}_2^\pi)(\mathcal{T}) \leq 2\sum_{t\in[H]} C(\psi_\xi;\pi)\cdot\mathrm{TV}(\mathbb{P}_1^{\psi_\xi},\mathbb{P}_2^{\psi_\xi})(\mathcal{T})$$

$$\leq 2\sum_{t\in[H]}\sum_{\psi\in\Psi_\xi}\frac{C(\psi_\xi;\pi)}{|\Psi_\xi|}\cdot\mathrm{TV}(\mathbb{P}_1^\psi,\mathbb{P}_2^\psi)(\mathcal{T})$$

$$\leq 2H\cdot\left(\frac{C(\psi_\xi;\pi)}{\sqrt{|\Psi_\xi|}}\right)\sqrt{\sum_{j=0}^{k-1}\mathrm{TV}^2(\mathbb{P}_1^{\pi^j},\mathbb{P}_2^{\pi^j})(\mathcal{T})}. \tag{16}$$

Then we apply Lemma A.4 and Lemma A.5 to deduce that

$$\sum_{j=0}^{k-1}\mathrm{TV}^2(\mathbb{P}_1^{\pi^j},\mathbb{P}_2^{\pi^j})(\mathcal{T}) \leq \frac{16\beta}{n_{\text{test}}}.$$

On the other hand, note that

$$C(\psi_\xi;\pi^k) = \max_{t\in[H]}\max_{x\in\mathcal{X}}\frac{\mathbb{P}^{\pi^k}(x_t = x)}{\mathbb{P}^{\psi_\xi}(x_t = x)} \leq \max_{x\in\mathcal{S}\times\mathcal{A}}\frac{|\Psi_\xi|\cdot\mathbb{P}^{\pi^k}(x_t = x)}{\max_{j<k}\mathbb{P}^{\pi^j}(x_t = x)}.$$

Now using the while loop condition, we have

$$\epsilon_{\text{TV}} < \mathrm{TV}(\mathbb{P}_1^{\pi^k},\mathbb{P}_2^{\pi^k})(\mathcal{T}) \leq 8H\cdot\max_{t\in[H]}\frac{C(\psi_\xi;\pi^k)}{\sqrt{|\Psi_\xi|}}\sqrt{n_{\text{test}}\beta}$$

$$\leq 8H\cdot\sqrt{\frac{HSA\beta}{n_{\text{test}}}}\max_{t\in[H]}\max_{x\in\mathcal{X}}\frac{\mathbb{P}^{\pi^k}(x_t = x)}{\max_{j<k}\mathbb{P}^{\pi^j}(x_t = x)}.$$

Rearranging the inequality, implies that there exists a $t \in [H]$ and an $x \in \mathcal{X}$ such that

$$\max_{j<k}\mathbb{P}^{\pi^j}(x_t = x) \leq \frac{8H}{\epsilon_{\text{TV}}}\sqrt{\frac{HSA\beta}{n_{\text{test}}}}\cdot\mathbb{P}^{\pi^k}(x_t = x).$$

## C.3 Proof of Theorem 3.3

We first show that Algorithm 1 terminates after $K = HSA\log(1/\gamma)$ iterations where $\gamma = \epsilon_{\text{test}}^2/H^2$. We consider a perturbed model $\hat{\theta}^* = (w, \hat{T}, R)$ where

$$\hat{T}(\cdot|s,a) = (1-\gamma)T^*(\cdot|s,a) + \gamma\mathbb{1},$$

By simulation lemma [38], for any $\pi \in \Pi$, note that $\mathrm{TV}(\mathbb{P}_1^\pi,\mathbb{P}_2^\pi)(y|x) \leq 2\gamma S$ for all $x, y$, and thus

$$\mathrm{TV}(\mathbb{P}_{\hat{\theta}^*}^\pi,\mathbb{P}_{\theta^*}^\pi)(\mathcal{T}) = \sum_{x_{1:H}}\sum_{y_{1:H}}\pi_{1:H}\left|\prod_{t=0}^H\mathbb{P}_1(y_t|x_t) - \prod_{t=0}^H\mathbb{P}_2(y_t|x_t)\right|$$

$$\leq \sum_{x_{1:H-1}}\sum_{y_{1:H-1}}\pi_{1:H-1}\left|\prod_{t=0}^{H-1}\mathbb{P}_1(y_t|x_t) - \prod_{t=0}^{H-1}\mathbb{P}_2(y_t|x_t)\right|$$

$$+ \sum_{x_H}\sum_{y_H}\mathbb{P}_1^\pi(x_H)|\mathbb{P}_1(y_H|x_H) - \mathbb{P}_2(y_H|x_H)|$$

$$\leq \sum_{x_{1:H-1}} \sum_{y_{1:H-1}} \pi_{1:H-1} \left| \prod_{t=0}^{H-1} \mathbb{P}_1(y_t|x_t) - \prod_{t=0}^{H-1} \mathbb{P}_2(y_t|x_t) \right| + 2\gamma S$$

$$\leq \dots \leq 2S\gamma H.$$

For an arbitrary $k$ iteration, we check whether the coverage doubling argument (Lemma D.1) still holds. To see this, first note that we can define $\hat{\Psi}_\xi$, $\hat{\psi}_\xi$ and $\hat{C}(\hat{\psi}_\xi; \pi)$ as in Lemma 3.2 in terms of $\hat{\theta}^*$:

$$\hat{\Psi}_\xi := \left\{ \pi^{j^*(x,t)}, \forall x, t \mid j^*(x,t) := \arg\max_{j \in 0,1,\dots,k-1} \mathbb{P}_{\hat{\theta}^*}^{\pi^j}(x_t = x), \right\},$$

and $\hat{\psi}_\xi$ is a checkpoint-dependent policy where $\hat{\psi}_\xi$ is a mixture of $\hat{\Psi}_\xi$, and

$$\hat{C}(\psi; \pi) := \max_{t \in [H]} \max_{x \in \mathcal{X}} \frac{\mathbb{P}_{\hat{\theta}^*}^\pi(x_t = x)}{\mathbb{P}_{\hat{\theta}^*}^\psi(x_t = x)}.$$

Now we invoke Lemma A.4 and Lemma A.5 to show that

$$\mathrm{TV}\left(\mathbb{P}_{\hat{\theta}^*}^{\hat{\psi}_\xi}, \mathbb{P}_\theta^{\hat{\psi}_\xi}\right)(\mathcal{T}) \leq \mathrm{TV}\left(\mathbb{P}_{\hat{\theta}^*}^{\hat{\psi}_\xi}, \mathbb{P}_{\theta^*}^{\hat{\psi}_\xi}\right)(\mathcal{T}) + \mathrm{TV}\left(\mathbb{P}_{\theta^*}^{\hat{\psi}_\xi}, \mathbb{P}_\theta^{\hat{\psi}_\xi}\right)(\mathcal{T})$$

$$\leq 2SH\gamma + \frac{1}{|\hat{\Psi}_\xi|} \sum_{\psi \in \hat{\Psi}_\xi} \mathrm{TV}\left(\mathbb{P}_{\theta^*}^\psi, \mathbb{P}_\theta^\psi\right)(\mathcal{T}),$$

for all $\theta \in \mathcal{C}^k$ using the triangle inequality for TV distance and $(a+b)^2 \leq 2(a^2+b^2)$. Thus, we can derive equation (16) in terms of $\hat{\theta}^*$ as

$$\mathrm{TV}(\mathbb{P}_{\hat{\theta}^*}^\pi, \mathbb{P}_\theta^\pi)(\mathcal{T}) \leq 2H \cdot \frac{\hat{C}(\hat{\psi}_\xi; \pi)}{\sqrt{|\hat{\Psi}_\xi|}} \sqrt{\frac{16\beta}{n_{\mathtt{test}}} + (2SH)^2\gamma^2}$$

$$\leq 2H\sqrt{HSA}\sqrt{\frac{16\beta}{n_{\mathtt{test}}} + (2SH)^2\gamma^2} \cdot \max_{t \in [H]} \max_{x \in \mathcal{X}} \frac{\mathbb{P}_{\hat{\theta}^*}^{\pi^k}(x_t = x)}{\max_{j<k} \mathbb{P}_{\hat{\theta}^*}^\psi(x_t = x)}$$

On the other hand,

$$\max\left(\mathrm{TV}(\mathbb{P}_{\hat{\theta}^*}^{\pi_k}, \mathbb{P}_{\theta_1}^{\pi_k})(\mathcal{T}), \mathrm{TV}(\mathbb{P}_{\hat{\theta}^*}^{\pi_k}, \mathbb{P}_{\theta_2}^{\pi_k})(\mathcal{T})\right) > 2\epsilon_{\mathtt{TV}} - 2SH\gamma > 1.5\epsilon_{\mathtt{TV}},$$

by setting $\gamma = \epsilon_{\mathtt{test}}/(4SAH)^4$. Let $n_{\mathtt{test}} = 64\beta(H^3SA)/\epsilon_{\mathtt{test}}^2$, and we have

$$2\epsilon_{\mathtt{TV}} < \epsilon_{\mathtt{test}} \cdot \max_{t \in [H]} \max_{x \in \mathcal{X}} \frac{\mathbb{P}_{\hat{\theta}^*}^\pi(x_t = x)}{\max_{j<k} \mathbb{P}_{\hat{\theta}^*}^\psi(x_t = x)}.$$

Hence the same doubling argument holds, and `MDP-OMLE` (Algorithm 1) will terminate after at most

$$K = O(HSA\log(HSA/\epsilon_{\mathtt{test}}))$$

iterations. We note that all inequalities hold for all $K$ iterations with probability at least $1 - \eta$. Finally, by setting $\epsilon_{\mathtt{test}} = \epsilon_{\mathtt{TV}}$ and $\epsilon_{\mathtt{TV}} = \epsilon/H$, we can conclude that the total number of trajectories that was generated by `MDP-OMLE` is bounded by

$$O(1) \cdot H^6 S^2 A^2 \beta \log(HSA/\epsilon)/\epsilon^2.$$

## Appendix D    Proofs for Section 4

### D.1    Proof Sketch

In this section, we provide an overview of the proof for Theorem 4.5. Compared to Algorithm 1, the main differences in LMDPs from the MDP cases are two-fold:

(a)  Our goal is to find the optimal *history-dependent* (non-Markovian) policy.

(b) We cannot observe any context-segment pair that previous behavioral policies could not cover.

For the first point, (a), we already have reduced the problem from matching all history-dependent policies to a set of behavioral policies generated by the concatenation of memoryless policies in Lemma 4.4. For the second point, (b), even though we cannot observe the latent context $m$ under which each segment is covered, we can improve the coverage of each context-segment pair given the test set $\Psi_{\text{test}}^k$ every iteration:

**Lemma D.1** *Let* $n_{test} \geq 3\beta M^2 (8H^2)^d (MS^2 A^2)^d / \epsilon_{test}^2$. *Then with probability at least* $1 - \eta$, *at every* $k^{th}$ *iteration in Algorithm 2, there must exist at least one* $(m, t_1, t_2, s, s')$, *such that*

$$P_m^{\pi^k}(s_{t_2} = s | s_{t_1}' = s') > 2 \cdot \max_{\psi \in \Psi_{test}^{k-1}} P_m^\psi(s_{t_2} = s | s_{t_1}' = s').$$

That is, we can ensure that the coverage of at least one context-segment pair is being exponentially improved despite the unobservability of latent contexts $m$.

For a moment, to simplify the discussion, we first assume that the uniformly random policy $\text{Unif}(A)$ has a non-zero $\gamma > 0$ probability for covering all segments in all contexts:

$$P_m^{\text{Unif}(A)}(s_{t_2} = s_2 | s_{t_1} = s_1) \geq \gamma, \qquad \forall t_1 < t_2, m, s_1, s_2. \tag{17}$$

We note that this assumption will be eventually removed in our final result. Thus, if we start from the initial coverage $\gamma > 0$ over all context-segment pairs, then since every probability is in the range of $[0, 1]$, this doubling-up event can happen at most $\log(1/\gamma)$ times for every context-segment pair. Therefore, Algorithm 2 must terminate after at most $K = MS^2 H \log(1/\gamma)$ iterations.

Separately from the coverage improvement in Lemma D.1, the standard concentration of the confidence set on the generated trajectory data is given by the maximum-likelihood estimation:

**Lemma D.2** *With probability at least* $1 - \eta$, *for all* $k^{th}$ *iterations in Algorithm 2, let* $\Psi_\xi = \{\psi_1, ..., \psi_n\} \subseteq \Psi_{test}^{k-1}$ *be any subset of candidate test policies, and let* $\psi_\xi = (\psi_\xi, ..., \psi_\xi, \text{Unif}(\mathcal{A}))$ *where* $\psi_\xi := \frac{1}{n} \sum_{i \in [n]} \psi_i$ *is a mixture of policies in* $\Psi_\xi$. *Then for any model in the confidence set* $\theta \in \mathcal{C}$, *the following holds:*

$$\sum_{\tau \in SubSeq(H,d)} \sum_{z \subseteq \{0,1\}^{\otimes |\tau|}} TV^2 \left( \mathbb{P}_{\theta^*}^{\nu(\psi_\xi; \tau, z)}, \mathbb{P}_{\theta}^{\nu(\psi_\xi; \tau, z)} \right) (x_\tau, y_\tau) \leq \frac{4\beta}{n^d \cdot n_{test}}.$$

Equipped with Lemma D.2 With probability at least $1 - \eta$, we terminate the while-loop after at most $K = HS^2 M \log(1/\gamma)$ iterations, and each while-loop generates $K^{d-1} \cdot n_{\text{test}}$ new trajectories, leading to a total

$$K^d \cdot n_{\text{test}} = (8M^2 S^4 H^3 A^2 \log(1/\gamma))^d \cdot O(M^2 \beta / \epsilon_{\text{test}}^2)$$

sample complexity. The near-optimality guarantee for the final empirical model is given by Lemma 4.4 where we set $\epsilon_{\text{test}} = M^{-1}(2H^2 MSA)^{-d} \cdot \epsilon_{\text{TV}}$ to obtain an $(H\epsilon_{\text{TV}})$-optimal policy.

**Without Initial Coverage.** Now we remove the initial $\gamma > 0$ coverage assumption (17). To do so, consider a virtual LMDP model $\hat{\theta} = (\{w_m^*, \hat{T}_m, R_m^*\})_{m=1}^M$ with perturbed transition models, *i.e.*, $\hat{T}_m(\cdot|s, a) = (1 - \gamma)T_m^*(\cdot|s, a) + \gamma \mathbb{1}$ for all $(s, a) \in \mathcal{S} \times \mathcal{A}$. For $\hat{\theta}$, it is easy to see that for all $\pi \in \Pi$, we have $\text{TV}(\mathbb{P}_{\hat{\theta}}^\pi, \mathbb{P}_{\theta^*}^\pi)(\tau) \leq 2HS\gamma$. Thus, now we can shift our arguments to this perturbed model with sufficiently small $\gamma$, and it is straightforward that the perturbed model has the $\gamma > 0$ segment coverage for any policy, which concludes Theorem 4.5.

## D.2   Proof of Lemma 4.2

We start from equation (12) in Lemma A.1. We refer the reader to Appendix A.2 for the used notation here, with additional notation we define here: for a subset of indices $I \subseteq [|\tau|]$, we write $\tau_I := (\tau_i)_{i \in I}$ to refer to a subsequence of $\tau$ at positions $I$, and $\tau_{/I}$ a subsequence at positions outside of $I$.

We can first bound $\Delta(x_{\boldsymbol{\tau}}, y_{\boldsymbol{\tau}})$ as the following:

$$\Delta(x_{\boldsymbol{\tau}}, y_{\boldsymbol{\tau}}) \leq \sum_{I \subseteq [|\boldsymbol{\tau}|]} \left( \prod_{i \in I} l_{\tau_i}^{i-1}(x_{\tau_{1:i-1}}, y_{\tau_{1:i-1}}) \right) \times$$

$$\left| \sum_m w_m^1 \left( \prod_{i \in I} P_m^{1,\nu(\psi_{i-1};\tau_{i-1})}(s_{\tau_i+1}|s_{\tau_{i-1}+1}) \right) \left( \prod_{i \in [|\boldsymbol{\tau}|]/I} P_m^{1,\nu(\psi_{i-1};\tau_{i-1})}(s_{\tau_i}|s_{\tau_{i-1}+1}) P_m^1(y_{\tau_i}|x_{\tau_i}) \right) \right.$$

$$\left. - \sum_m w_m^2 \left( \prod_{i \in I} P_m^{2,\nu(\psi_{i-1};\tau_{i-1})}(s_{\tau_i+1}|s_{\tau_{i-1}+1}) \right) \left( \prod_{i \in [|\boldsymbol{\tau}|]/I} P_m^{2,\nu(\psi_{i-1};\tau_{i-1})}(s_{\tau_i}|s_{\tau_{i-1}+1}) P_m^2(y_{\tau_i}|x_{\tau_i}) \right) \right|$$

$$= \sum_{I \subseteq [|\boldsymbol{\tau}|]} \frac{\prod_{i \in I} l_{\tau_i}^{i-1}(x_{\tau_{1:i-1}}, y_{\tau_{1:i-1}})}{\prod_{i \in [\boldsymbol{\tau}]/I}(1/A)} \delta_{\nu(\boldsymbol{\psi};\boldsymbol{\tau}, z(I;\boldsymbol{\tau}))}(s'_{\boldsymbol{\tau}_I}, x_{\boldsymbol{\tau}_{/I}}, y_{\boldsymbol{\tau}_{/I}}),$$

where $z(I; \boldsymbol{\tau})$ satisfies $z(I; \boldsymbol{\tau})_j = 0$ if $j \in I$ and 1 otherwise. Then we can observe that

$$\frac{\prod_{i \in I} l_{\tau_i}^{i-1}(x_{\tau_{1:i-1}}, y_{\tau_{1:i-1}})}{\prod_{i \in [q]/I}(1/A)} \cdot \frac{1}{\prod_{i=0}^{q-1} P_{m(x_{\boldsymbol{\tau}}, y_{\boldsymbol{\tau}})}^{\nu(\psi_i;\tau_i)}(s_{\tau_i+1}|s_{\tau_i+1}) P_{m(x_{\boldsymbol{\tau}}, y_{\boldsymbol{\tau}})}(y_{\tau_i+1}|x_{\tau_i+1})}$$

$$\leq \frac{1}{\prod_{i \in I} P_{m(x_{\boldsymbol{\tau}}, y_{\boldsymbol{\tau}})}^{\nu(\psi_{i-1};\tau_{i-1})}(s_{\tau_i+1}|s_{\tau_{i-1}+1}) \cdot \prod_{i \in [q]/I} P_{m(x_{\boldsymbol{\tau}}, y_{\boldsymbol{\tau}})}^{\nu(\psi_{i-1};\tau_{i-1})}(x_{\tau_i}|s_{\tau_{i-1}+1}) P_{m(x_{\boldsymbol{\tau}}, y_{\boldsymbol{\tau}})}(y_{\tau_i}|x_{\tau_i})}$$

$$= \frac{1}{P_{m(x_{\boldsymbol{\tau}}, y_{\boldsymbol{\tau}})}^{\nu(\boldsymbol{\psi};\boldsymbol{\tau}, z(I;\boldsymbol{\tau}))}(s'_{\boldsymbol{\tau}_I}, x_{\boldsymbol{\tau}_{/I}}, y_{\boldsymbol{\tau}_{/I}})}, \tag{18}$$

using inequality that can be derived by definition of $l_t^i$:

$$l_{\tau_i}^{i-1} \leq \frac{P_{m(x_{\boldsymbol{\tau}}, y_{\boldsymbol{\tau}})}^{\nu(\psi_{i-1};\tau_{i-1})}(s_{\tau_i}|s_{\tau_{i-1}+1})}{P_{m(x_{\boldsymbol{\tau}}, y_{\boldsymbol{\tau}})}^{\nu(\psi_{i-1};\tau_{i-1})}(s_{\tau_i+1}|s_{\tau_{i-1}+1})} P_{m(x_{\boldsymbol{\tau}}, y_{\boldsymbol{\tau}})}(y_{\tau_i}|x_{\tau_i}).$$

We are now ready to use this inequality to bound the TV distance between trajectory distribution via equation (12). To do so, we rearrange the summation orders as follows:

$$\sum_{x_{1:H}} \sum_{y_{1:H}} \pi_{1:H} \left| \sum_{m=1}^{M_1} w_m^1 \prod_{t=0}^H P_m^1(y_t|x_t) - \sum_{m=1}^{M_2} w_m^2 \prod_{t=0}^H P_m^2(y_t|x_t) \right|$$

$$\leq \sum_{\boldsymbol{\tau} \subseteq [H]} \sum_{m \in [M_1]} \sum_{x_{\boldsymbol{\tau}}, y_{\boldsymbol{\tau}}:m(x_{\boldsymbol{\tau}}, y_{\boldsymbol{\tau}})=m} \Delta(x_{\boldsymbol{\tau}}, y_{\boldsymbol{\tau}}) \times \left( \frac{P_m^\pi(x_{\boldsymbol{\tau}}, y_{\boldsymbol{\tau}})}{\prod_{i=0}^{q-1} P_m^{\nu(\psi_i;\tau_i)}(s_{\tau_i+1}|s_{\tau_i+1}) P_m(y_{\tau_i+1}|x_{\tau_i+1})} \right)$$

$$\leq \sum_{\boldsymbol{\tau} \subseteq [H]} \sum_{m \in [M_1]} \sum_{x_{\boldsymbol{\tau}}, y_{\boldsymbol{\tau}}:m(x_{\boldsymbol{\tau}}, y_{\boldsymbol{\tau}})=m} \sum_{I \subseteq [q]} \left( \frac{P_m^\pi(x_{\boldsymbol{\tau}}, y_{\boldsymbol{\tau}}) \cdot \delta_{\nu(\boldsymbol{\psi};\boldsymbol{\tau}, z(I;\boldsymbol{\tau}))}(s'_{\boldsymbol{\tau}_I}, x_{\boldsymbol{\tau}_{/I}}, y_{t_{\boldsymbol{\tau}_{/I}}})}{P_m^{\nu(\boldsymbol{\psi};\boldsymbol{\tau}, z(I;\boldsymbol{\tau}))}(s'_{\boldsymbol{\tau}_I}, x_{\boldsymbol{\tau}_{/I}}, y_{\boldsymbol{\tau}_{/I}})} \right)$$

$$\leq \sum_{\boldsymbol{\tau} \subseteq [H]} \sum_{m \in [M_1]} \sum_{I \subseteq [q]} \sum_{s'_{\boldsymbol{\tau}_I}, x_{\boldsymbol{\tau}_{/I}}, y_{\boldsymbol{\tau}_{/I}}} \left( \frac{P_m^\pi(s'_{\boldsymbol{\tau}_I}, x_{\boldsymbol{\tau}_{/I}}, y_{\boldsymbol{\tau}_{/I}}) \cdot \delta_{\nu(\boldsymbol{\psi};\boldsymbol{\tau}, z(I;\boldsymbol{\tau}))}(s'_{\boldsymbol{\tau}_I}, x_{\boldsymbol{\tau}_{/I}}, y_{\boldsymbol{\tau}_{/I}})}{P_m^{\nu(\boldsymbol{\psi};\boldsymbol{\tau}, z(I;\boldsymbol{\tau}))}(s'_{\boldsymbol{\tau}_I}, x_{\boldsymbol{\tau}_{/I}}, y_{\boldsymbol{\tau}_{/I}})} \right)$$

$$\leq M \cdot C(\boldsymbol{\psi}; \pi) \cdot \sum_{q \leq d} \sum_{\boldsymbol{\tau} \subseteq [H], |\boldsymbol{\tau}|=q} \sum_{I \subseteq [q]} \text{TV} \left( \mathbb{P}_1^{\nu(\boldsymbol{\psi};\boldsymbol{\tau}, \boldsymbol{z}(I;\tau))}, \mathbb{P}_2^{\nu(\boldsymbol{\psi};\boldsymbol{\tau}, \boldsymbol{z}(I;\tau))} \right)(s'_{\boldsymbol{\tau}_I}, x_{\boldsymbol{\tau}_{/I}}, y_{\boldsymbol{\tau}_{/I}})$$

$$\leq M \cdot C(\boldsymbol{\psi}; \pi) \cdot \sum_{q \leq d} \sum_{\boldsymbol{\tau} \subseteq [H], |\boldsymbol{\tau}|=q} \sum_{I \subseteq [q]} \text{TV} \left( \mathbb{P}_1^{\nu(\boldsymbol{\psi};\boldsymbol{\tau}, \boldsymbol{z}(I;\tau))}, \mathbb{P}_2^{\nu(\boldsymbol{\psi};\boldsymbol{\tau}, \boldsymbol{z}(I;\tau))} \right)(x_{\boldsymbol{\tau}}, y_{\boldsymbol{\tau}}),$$

proving Lemma 4.2.

### D.3 Proof of Lemma 4.4

Before proceeding to the proof, we refer the reader to Appendix A.3 for additional preliminaries. Recall the definition of $\Psi_\xi$ in Lemma A.3:

$$\Psi_\xi = \left\{ \arg \max_{\psi \in \Psi_{\text{test}}} P_m^{\nu(\psi;t_0)}(s_{t_0+t} = s | s'_{t_0} = s'), \forall m, s, s', t \right\} \subseteq \Psi_{\text{test}}.$$

Note that the above definition is invariant to $t_0$.

Now consider $\Psi_{\texttt{test}} = \Pi_{\text{mls}}$. Then, $\Psi_\xi$ is the set of policies that maximize the probability $P_m^{\nu(\psi;t_0)}(s_{t_0+t} = s|s'_{t_0} = s')$, *i.e.*, a memoryless policy that aims to reach $s$ after $t$ time steps (this policy is invariant to the starting state $s'$). Therefore, we can first induce that $|\Psi_\xi| \le MHS$. Then, the Markovian policy maximizing the probability to reach a certain state under a fixed context $m$ is also best within the history-dependent policy class $\Pi$, and therefore

$$\rho(\Pi_{\text{mls}}; \pi) := \max_{t,t_0} \max_{m,s,s'} \frac{\max_{\mathcal{T}_{1:t_0}} P_m^\pi(s_{t+t_0} = s|s'_{t_0} = s', \mathcal{T}_{1:t_0})}{\max_{\psi \in \Psi_\xi} P_m^{\nu(\psi;t_0)}(s_{t+t_0} = s|s'_{t_0} = s')} \le 1.$$

We can now invoke Lemma 4.2 and Lemma A.3, and noting that

$$\text{TV}(\mathbb{P}_1^\pi, \mathbb{P}_2^\pi)(s'_{\boldsymbol{\tau}_I}, x_{\boldsymbol{\tau}_{/I}}, y_{\boldsymbol{\tau}_{/I}}) \le \text{TV}(\mathbb{P}_1^\pi, \mathbb{P}_2^\pi)(\mathcal{T}).$$

Let $\boldsymbol{\psi}_\xi$ be as defined in Lemma A.3, and for any $\pi \in \Pi$, we can conclude that

$$\text{TV}(\mathbb{P}_1^\pi, \mathbb{P}_2^\pi)(\mathcal{T}) \le M \cdot C(\boldsymbol{\psi}_\xi; \pi) \sum_{\boldsymbol{\tau} \in \texttt{Subseq}(H,d)} \sum_{I \in [|\boldsymbol{\tau}|]} \text{TV}\left(\mathbb{P}_1^{\nu(\boldsymbol{\psi}_\xi; \boldsymbol{\tau}, z(I;\boldsymbol{\tau}))}, \mathbb{P}_2^{\nu(\boldsymbol{\psi}_\xi; \boldsymbol{\tau}, z(I;\boldsymbol{\tau}))}\right)(s'_{\boldsymbol{\tau}_I}, x_{\boldsymbol{\tau}_{/I}}, y_{\boldsymbol{\tau}_{/I}})$$

$$\le M \cdot (MHSA)^d \sum_{\boldsymbol{\tau} \in \texttt{Subseq}(H,d)} \sum_{\boldsymbol{z} \in \{0,1\}^{|\boldsymbol{\tau}|}} \text{TV}\left(\mathbb{P}_1^{\nu(\boldsymbol{\psi}_\xi; \boldsymbol{\tau}, \boldsymbol{z})}, \mathbb{P}_2^{\nu(\boldsymbol{\psi}_\xi; \boldsymbol{\tau}, \boldsymbol{z})}\right)(\mathcal{T}).$$

Finally, it only remains to bound the total variation distance with $\nu(\boldsymbol{\psi}_\xi; \boldsymbol{\tau}, \boldsymbol{z})$. To see this, for a given $q \le d$, $\boldsymbol{\tau} \in \texttt{Subseq}(H,d)$, and $\boldsymbol{z} \in \{0,1\}^{|\boldsymbol{\tau}|}$, it is easy to check that

$$\text{TV}\left(\mathbb{P}_1^{\nu(\boldsymbol{\psi}_\xi, \boldsymbol{\tau}, \boldsymbol{z})}, \mathbb{P}_2^{\nu(\boldsymbol{\psi}_\xi; \boldsymbol{\tau}, \boldsymbol{z})}\right)(\mathcal{T}) \le \max_{\boldsymbol{\psi} \in \Pi_{\text{mls}}^{\otimes(q+1)}} \text{TV}\left(\mathbb{P}_1^{\nu(\boldsymbol{\psi}; \boldsymbol{\tau}, \boldsymbol{z})}, \mathbb{P}_2^{\nu(\boldsymbol{\psi}; \boldsymbol{\tau}, \boldsymbol{z})}\right)(\mathcal{T}) \le \epsilon_{\texttt{test}}.$$

Noting that $\sum_{q \le d} \binom{H}{q} \le \min(H^d, 2^H)$, assuming we are in the regime $H \gg d$, we conclude that

$$\text{TV}(\mathbb{P}_1^\pi, \mathbb{P}_2^\pi)(\mathcal{T}) \le M(MSA)^d \cdot (2H^2)^d \cdot \epsilon_{\texttt{test}},$$

concluding the proof.

### D.4 Proof of Theorem 4.5

We continue from the conclusion of Lemma D.1, and the remaining step is to ensure that `LMDP-OMLE` terminates after $K = MS^2H \log(1/\gamma)$ iterations where $\gamma = \epsilon_{\texttt{test}}^2/(H^{2d})$ without the initial coverage assumption (17). We consider a perturbed model $\hat{\theta}^* = (\{w_m^*, \hat{T}_m, R_m^*\}_{m=1}^M)$ where

$$\hat{T}_m(\cdot|s,a) = (1-\gamma)T_m^*(\cdot|s,a) + \gamma\mathbb{1},$$

for all $m$. By the performance difference lemma [35], for any $\pi \in \Pi$,

$$\text{TV}(\mathbb{P}_{\hat{\theta}^*}^\pi, \mathbb{P}_{\theta^*}^\pi)(\mathcal{T}) \le \sum_m w_m^* \cdot \text{TV}(\mathbb{P}_{\hat{\theta}^*}^\pi, \mathbb{P}_{\theta^*}^\pi)(\mathcal{T}|m) \le 2S\gamma H.$$

For every $k^{th}$ iteration, we check whether the coverage doubling argument (Lemma D.1) still holds. To see this, first note that we can define $\hat{\rho}(\Psi_{\texttt{test}}^k; \pi)$ and $\hat{\Psi}_\xi$, $\hat{\psi}_\xi$ and $\hat{C}(\boldsymbol{\psi}_\xi; \pi)$ as in Lemma A.3 in terms of $\hat{\theta}^*$. Then Lemma D.2 can be modified to guarantee that

$$\sum_{\boldsymbol{\tau} \in \texttt{SubSeq}(H,d)} \sum_{\boldsymbol{z} \in \{0,1\}^{|\boldsymbol{\tau}|}} \text{TV}^2\left(\mathbb{P}_{\hat{\theta}^*}^{\nu(\hat{\psi}_\xi; \boldsymbol{\tau}, \boldsymbol{z})}, \mathbb{P}_\theta^{\nu(\hat{\psi}_\xi; \boldsymbol{\tau}, \boldsymbol{z})}\right)(\mathcal{T}) \le \frac{16\beta}{n^d \cdot n_{\texttt{test}}} + 2(2H)^d(2SH)^2\gamma^2,$$

for all $\theta \in \mathcal{C}^k$ using the triangle inequality for TV distance and $(a+b)^2 \le 2(a^2 + b^2)$. Thus, we can derive (19) in terms of $\hat{\theta}^*$ as

$$\text{TV}(\mathbb{P}_{\hat{\theta}^*}^\pi, \mathbb{P}_\theta^\pi)(\mathcal{T}) \le 8M(nA\hat{\rho}(\hat{\Psi}_\xi; \pi))^d\sqrt{\frac{(2H)^d\beta}{n^d \cdot n_{\texttt{test}}} + (2H)^d(4SH)^2\gamma^2}.$$

| History (with Possible Contexts) | $a_2 = 1$ | $a_2 = 2$ |
|---|---|---|
| $a_1 = 1, r_1 = -1 \ (m = 1)$ | ? | $\mathbb{E}[r_2] = 1$ |
| $a_1 = 2, r_1 = 1 \ (m = 2)$ | $\mathbb{E}[r_2] = -1$ | ? |
| $a_1 = 2, r_1 \neq 1 \ (m = 1 \text{ or } 3)$ | $\mathbb{E}[r_2] = 0$ | ? |
| $a_1 = 1, r_1 \neq -1 \ (m = 2 \text{ or } 3)$ | ? | $\mathbb{E}[r_2] = 0$ |

**Table 1:** The first step generates one of four possible beliefs of a history. Measured elements in the table denote the expected rewards of actions at $t = 2$ following a behavioral policy. We can see that for all $m \in [M], a \in \mathcal{A}$ it holds that $\mathbb{P}_m(s_2, a_2) > 0$ for all $m \in [M], a_2 \in \mathcal{A}$; yet, we cannot estimate $\mathbb{E}[r_2]$ given some histories.

On the other hand,

$$\max\left(\text{TV}(\mathbb{P}_{\hat{\theta}^*}^{\pi_k}, \mathbb{P}_{\theta_1}^{\pi_k})(\mathcal{T}), \text{TV}(\mathbb{P}_{\hat{\theta}^*}^{\pi_k}, \mathbb{P}_{\theta_2}^{\pi_k})(\mathcal{T})\right) > 2\epsilon_{\text{TV}} - 2SH\gamma > 1.5\epsilon_{\text{TV}},$$

Now we set $\gamma = \epsilon_{\text{test}}^2 (8HnA)^{-2d}(4MSH)^{-2}$ with $n_{\text{test}} = 64M^2\beta(HnA^2)^d/\epsilon_{\text{test}}^2$ and $n = MHS^2$, and we have

$$2\epsilon_{\text{TV}}^2 < 4^{-d} \cdot \hat{\rho}(\Psi_{\text{test}}^{k-1}; \pi^k)^d \epsilon_{\text{test}}^2.$$

Hence the same doubling argument holds, and Algorithm 2 will terminate after at most

$$K = O(MdS^2 H \log(MHSA/\epsilon_{\text{test}}))$$

iterations. We note that all inequalities hold for all $K$ iterations with probability at least $1 - \eta$. Finally, we invoke Lemma 4.4 with $\epsilon_{\text{test}} = \epsilon_{\text{TV}} \cdot \text{poly}(H, M, S, A)^{-d}$, $\epsilon_{\text{TV}} = \epsilon/(4MSAH)^d$ and $d = 2M - 1$, we can conclude that the total number of trajectories we generated is bounded by

$$\text{poly}(M, H, S, A, \log(MHSA/\epsilon))^d/\epsilon^2.$$

## D.5 A Counter Example for Remark 4.3

One may wonder why it is not sufficient to consider a single latent-state coverability analogous to Lemma 3.1, analogous to equation (2), defined as the following:

$$\max_{t \in [H]} \max_{x \in \mathcal{X}} \max_{m \in [M]} \frac{P_m^\pi(x_t = x)}{P_m^\psi(x_t = x)}.$$

Here we present a counter example where the multi-step events must be considered even in the latent state space: the LMDP consists of $M = 3$ MDPs with $\mathcal{S} = \{1, 2\}, \mathcal{A} = \{1, 2\}, \mathcal{R} = \{-1, 0, 1\}$, and $H = 2$. All MDP starts from $s_1 = 1$ and with a transition kernel $T_m(s_2 = 2|s_1, a) = 1$ for all $m \in [M]$ and $a \in \mathcal{A}$. Reward models are given by $R_1(r = -1|s = 1, a = 1) = 1$, $R_2(r = 1|s = 1, a = 2) = 1$, and $R_m(r = 1|s = 1, a = 1) = 0.5$ for $r \in \{0, 1\}$, $m \in \{2, 3\}$, and $R_m(r|s = 1, a = 2) = 0.5$ for $r \in \{-1, 0\}$, $m \in \{1, 3\}$.

Now we target to measure the expected rewards of actions executed by a behavioral policy at $s_2 = 2$ as in Table 1. In this example, all actions are covered under all contexts following the behavior policy, *i.e.,* $P_m(s_2, a_2) > 0$. Yet, we cannot estimate the expected reward of action $a_2 = 1$ under context $m = 1$. Therefore, the speculated single latent-state coverage coefficient is finite for this problem, however, off-policy evaluation guarantee cannot be established only with the single latent-state coverability.

# Appendix E  Deferred Proofs

## E.1 Proof of Lemma D.1

For any fixed checkpoint $t_0$, recall the definition $\Psi_\xi$ as defined in equation (14):

$$\Psi_\xi = \left\{\arg\max_{\psi \in \Pi_{\text{test}}^{k-1}} P_m^{\nu(\psi; t_0)}(s_{t+t_0} = s|s'_{t_0} = s'), \ \forall m, s, s', t\right\}.$$

Note that $|\Psi_\xi| \leq MS^2H$ and invariant to $t_0$ since $\Psi_{\text{test}}^{k-1} \subset \Pi_{\text{mls}}$. Now for any memoryless policy $\pi \in \Pi_{\text{mls}}$, we recall Lemma 4.2 with $\boldsymbol{\psi}_\xi = (\psi_\xi, ..., \psi_\xi, \text{Unif}(\mathcal{A}))$, where $\psi_\xi \in \Pi$ is a mixture policy of $\Psi_\xi$. We have

$$\text{TV}(\mathbb{P}_{\theta^*}^\pi, \mathbb{P}_\theta^\pi)(\mathcal{T}) \leq MC(\boldsymbol{\psi}_\xi; \pi) \sum_{\boldsymbol{\tau} \in \text{Subseq}(H, d)} \sum_{\boldsymbol{z} \in \{0,1\}^{|\tau|}} \text{TV}\left(\mathbb{P}_{\theta^*}^{\nu(\boldsymbol{\psi}; \boldsymbol{\tau}, \boldsymbol{z})}, \mathbb{P}_\theta^{\nu(\boldsymbol{\psi}; \boldsymbol{\tau}, \boldsymbol{z})}\right)(\mathcal{T})$$

$$\leq MC(\boldsymbol{\psi}_\xi; \pi)\sqrt{(2H)^d} \cdot \sqrt{\sum_{\boldsymbol{\tau} \in \mathtt{Subseq}(H,d)} \sum_{\boldsymbol{z} \in \{0,1\}^{|\boldsymbol{\tau}|}} \mathtt{TV}^2\left(\mathbb{P}_{\theta^*}^{\nu(\boldsymbol{\psi};\boldsymbol{\tau},\boldsymbol{z})}, \mathbb{P}_\theta^{\nu(\boldsymbol{\psi};\boldsymbol{\tau},\boldsymbol{z})}\right)(\mathcal{T})}$$

$$\leq 4M(nA\rho(\Psi_{\mathtt{test}}^{k-1}; \pi))^d \sqrt{\frac{(2H)^d\beta}{n^d \cdot n_{\mathtt{test}}}}, \tag{19}$$

where the last inequality follows by Lemma D.2. By the while-loop condition, for $\pi_k \in \Pi_{\mathrm{mls}}$, we have

$$\max\left(\mathtt{TV}(\mathbb{P}_{\theta^*}^{\pi_k}, \mathbb{P}_{\theta_1}^{\pi_k})(\mathcal{T}), \mathtt{TV}(\mathbb{P}_{\theta^*}^{\pi_k}, \mathbb{P}_{\theta_2}^{\pi_k})(\mathcal{T})\right) > 2\epsilon_{\mathtt{TV}},$$

by the triangle inequality for TV distance. Therefore, we can conclude that

$$4\epsilon_{\mathtt{TV}}^2 < M^2(A\rho(\Psi_{\mathtt{test}}^{k-1}; \pi^k))^{2d} \cdot \frac{16(2nH)^d\beta}{n_{\mathtt{test}}}.$$

Plugging $n_{\mathtt{test}} = 64M^2\beta(8HnA^2)^d/\epsilon_{\mathtt{test}}^2$ with $n = MHS^2$, we have

$$4^d < \rho(\Psi_{\mathtt{test}}^{k-1}; \pi^k)^{2d} = \rho(\Psi_{\mathtt{test}}^{k-1}; \pi^k)^{2d}.$$

Thus, $\rho(\Psi_{\mathtt{test}}^{k-1}; \pi^k) > 2$, which in turn implies Lemma D.1.

### E.2  Proof of Lemma D.2

By the construction of confidence set in equation (1) and Lemma A.5, for all $k \in [K]$ and $\theta \in \mathcal{C}_k$, we have

$$\sum_{(\mathcal{T}, \pi) \in \mathcal{D}_k} \mathtt{TV}^2(\mathbb{P}_{\theta^*}^\pi, \mathbb{P}_\theta^\pi)(\mathcal{T}) \leq 3\beta,$$

where $\beta = \log(K|\Theta|/\eta)$. Let $\psi_\xi$ be a mixture of a subset of candidate policies $\Psi_\xi = \{\psi_1, \psi_2, ..., \psi_n\} \subseteq \Psi_{\mathtt{test}}^{k-1}$. With $\boldsymbol{\psi}_\xi = (\psi_\xi, \psi_\xi, ..., \psi_\xi, \mathtt{Unif}(\mathcal{A}))$ and for every $\boldsymbol{\tau} \in \mathtt{Subseq}(H,d), \boldsymbol{z} \in \{0,1\}^{|\boldsymbol{\tau}|}$, we have

$$\mathtt{TV}\left(\mathbb{P}_{\theta^*}^{\nu(\boldsymbol{\psi}_\xi;\boldsymbol{\tau},\boldsymbol{z})}, \mathbb{P}_\theta^{\nu(\boldsymbol{\psi}_\xi;\boldsymbol{\tau},\boldsymbol{z})}\right)(\mathcal{T})$$

$$\leq \frac{1}{n^d} \sum_{i_1,i_2,...,i_d \in [n]} \mathtt{TV}\left(\mathbb{P}_{\theta^*}^{\nu((\psi_{i_1},...,\psi_{i_d},\mathtt{Unif}(\mathcal{A}));\boldsymbol{\tau},\boldsymbol{z})}, \mathbb{P}_\theta^{\nu((\psi_{i_1},...,\psi_{i_d},\mathtt{Unif}(\mathcal{A}));\boldsymbol{\tau},\boldsymbol{z})}\right)(\mathcal{T})$$

$$\leq \sqrt{\frac{1}{n^d}} \sqrt{\sum_{i_1,i_2,...,i_d \in [n]} \mathtt{TV}^2\left(\mathbb{P}_{\theta^*}^{\nu((\psi_{i_1},...,\psi_{i_d},\mathtt{Unif}(\mathcal{A}));\boldsymbol{\tau},\boldsymbol{z})}, \mathbb{P}_\theta^{(\nu(\psi_{i_1},...,\psi_{i_d},\mathtt{Unif}(\mathcal{A}));\boldsymbol{\tau},\boldsymbol{z})}\right)(\mathcal{T})}.$$

Therefore,

$$\sum_{\boldsymbol{\tau} \in \mathtt{Subseq}(H,d)} \sum_{I \subseteq [|t|]} \mathtt{TV}^2\left(\mathbb{P}_{\theta^*}^{\nu(\boldsymbol{\psi}_\xi;\boldsymbol{\tau},z(I;\boldsymbol{\tau}))}, \mathbb{P}_\theta^{\nu(\boldsymbol{\psi}_\xi;\boldsymbol{\tau},z(I;\boldsymbol{\tau}))}\right)(s'_{\boldsymbol{\tau}_I}, x_{\boldsymbol{\tau}/I}, y_{\boldsymbol{\tau}/I})$$

$$\leq \frac{1}{n^d} \sum_{\boldsymbol{\tau} \in \mathtt{Subseq}(H,d)} \sum_{\boldsymbol{z} \in \{0,1\}^{|\boldsymbol{\tau}|}} \sum_{i_1,i_2,...,i_d \in [n]} \mathtt{TV}^2\left(\mathbb{P}_{\theta^*}^{\nu((\psi_{i_1},...,\psi_{i_d},\mathtt{Unif}(\mathcal{A}));\boldsymbol{\tau},\boldsymbol{z})}, \mathbb{P}_\theta^{\nu((\psi_{i_1},...,\psi_{i_d},\mathtt{Unif}(\mathcal{A}));\boldsymbol{\tau},\boldsymbol{z})}\right)(\mathcal{T})$$

$$\leq \frac{3\beta}{n^d \cdot n_{\mathtt{test}}}.$$

where the last inequality is due to the construction of our dataset $\mathcal{D}^k$ and the concentration guarantee for the confidence set $\mathcal{C}^k$.

### E.3  Proof of Lemma A.1

First note that we can rewrite an *atomic* probability $P_m^n(y_t|x_t) = P_m^n(r_t, s_{t+1}|x_t)$ as

$$P_m^n(r_t, s_{t+1}|x_t) = \frac{P_m^{n,\psi_0}(s_{t+1})}{P_m^{n,\psi_0}(s_t)}\left(\frac{P_m^{n,\psi_0}(s_t)}{P_m^{n,\psi_0}(s_{t+1})}P_m^n(r_t, s_{t+1}|x_t) - l^0(x_t, r_t; s_{t+1}) + l^0(x_t, r_t; s_{t+1})\right),$$

In turn, moving from conditioning on the event $(x_t, y_t)$, we view the LMDP model after induction as if there are at most $M_1 - 1$ contexts in the first LMDP model (or $M_2 - 1$ in the second LMDP model if $l^0(x_t, r_t; s_t)$ is attained with $n = 2$). We often use a shorthand $l^0(x_t, y_t) = l^0(x_t, r_t; s_{t+1})$ to reduce the burden on the notation.

*Proof.* The basic strategy is to apply iteratively the triangle inequality. To illustrate, we first expand the equation at $t = 1$:

$$\sum_{x_{1:H}} \sum_{y_{1:H}} \pi_{1:H} \left| \sum_{m=1}^{M_1} w_m^1 \prod_{t=0}^{H} P_m^1(y_t|x_t) - \sum_{m=1}^{M_2} w_m^2 \prod_{t=0}^{H} P_m^2(y_t|x_t) \right|$$

$$= \sum_{x_1,y_1} \sum_{\substack{x_{2:H} \\ y_{2:H}}} \pi_{1:H} \left| \sum_{m=1}^{M_1} w_m^1 P_m^{1,\psi_0}(s_2) \left( \frac{P_m^{1,\psi_0}(s_1)}{P_m^{1,\psi_0}(s_2)} P_m^1(y_1|x_1) - l^0(x_1,y_1) + l^0(x_1,y_1) \right) \prod_{t=2}^{H} P_m^1(y_t|x_t) \right.$$

$$\left. - \sum_{m=1}^{M_2} w_m^2 P_m^{2,\psi_0}(s_2) \left( \frac{P_m^{2,\psi_0}(s_1)}{P_m^{2,\psi_0}(s_2)} P_m^2(y_1|x_1) - l^0(x_1,y_1) + l^0(x_1,y_1) \right) \prod_{t=2}^{H} P_m^2(y_t|x_t) \right|$$

$$\leq \sum_{x_1,y_1} \sum_{\substack{x_{2:H} \\ y_{2:H}}} \pi_{1:H} l^0(x_1,y_1) \left| \sum_{m=1}^{M_1} w_m^1 P_m^{1,\psi_0}(s_2) \prod_{t=2}^{H} P_m^1(y_t|x_t) - \sum_{m=1}^{M_2} w_m^2 P_m^{2,\psi_0}(s_2) \prod_{t=2}^{H} P_m^2(y_t|x_t) \right|$$

$$+ \sum_{x_1,y_1} \sum_{\substack{x_{2:H} \\ y_{2:H}}} \pi_{1:H} \left| \sum_{m=1}^{M_1} w_m^1 \left( P_m^{1,\psi_0}(s_1) P_m^1(y_1|x_1) - l^0(x_1,y_1) P_m^{1,\psi_0}(s_2) \right) \prod_{t=2}^{H} P_m^1(y_t|x_t) \right.$$

$$\left. - \sum_{m=1}^{M_2} w_m^2 \left( P_m^{2,\psi_0}(s_1) P_m^2(y_1|x_1) - l^0(x_1,y_1) P_m^{2,\psi_0}(s_2) \right) \prod_{t=2}^{H} P_m^2(y_t|x_t) \right|. \qquad (20)$$

We can continue applying triangle inequalities to all possible first event-logging time step, and we can start with the following inequality:

$$\sum_{x_{1:H}} \sum_{y_{1:H}} \pi_{1:H} \left| \sum_{m=1}^{M} w_m^1 \prod_{t=0}^{H} P_m^1(y_t|x_t) - \sum_{m=1}^{M} w_m^2 \prod_{t=0}^{H} P_m^2(y_t|x_t) \right|$$

$$\leq \sum_{\tau_1 \in [H]} \sum_{x_{1:\tau_1}, y_{1:\tau_1}} \pi_{1:\tau_1} l^0_{1:\tau_1-1} \times$$

$$\sum_{\substack{x_{\tau_1+1:H} \\ y_{\tau_1+1:H}}} \pi_{\tau_1+1:H} \left| \sum_{m=1}^{M} p_m^{1,1}(x_{\tau_1}, y_{\tau_1}) \prod_{t=\tau_1+1}^{H} P_m^1(y_t|x_t) - \sum_{m=1}^{M} p_m^{2,1}(x_{\tau_1}, y_{\tau_1}) \prod_{t=\tau_1+1}^{H} P_m^2(y_t|x_t) \right|.$$

Recall that at least one $p_m^{1,1}$ or $p_m^{2,1}$ is 0, that is, one of the contexts in either of the two systems is eliminated.

Now for $(i)$, we can pick the next event-logging time step $\tau_2 > \tau_1$, and apply the triangle inequality similarly, and repeat such event-logging until all contexts are exhausted. Since there are at most $2M$ contexts, we cannot repeat this process more than $d = 2M - 1$ times. Hence, we arrive to the following inequality:

$$\sum_{x_{1:H}} \sum_{y_{1:H}} \pi_{1:H} \left| \sum_{m=1}^{M} w_m^1 \prod_{t=0}^{H} P_m^1(y_t|x_t) - \sum_{m=1}^{M} w_m^2 \prod_{t=0}^{H} P_m^2(y_t|x_t) \right|$$

$$\leq \sum_{\boldsymbol{\tau} \in \mathrm{Subseq}(H,d)} \sum_{x_{\boldsymbol{\tau}}, y_{\boldsymbol{\tau}}} \Delta(x_{\boldsymbol{\tau}}, y_{\boldsymbol{\tau}}) \times \sum_{\substack{x_{0:\tau_1-1} \\ y_{0:\tau_1-1}}} \cdots \sum_{\substack{x_{\tau_{|\boldsymbol{\tau}|}+1:H} \\ y_{\tau_{|\boldsymbol{\tau}|}+1:H}}} \prod_{i=0}^{|\boldsymbol{\tau}|} \left( \pi_{\tau_i+1:\tau_{i+1}} \cdot l^i_{\tau_i+1:\tau_{i+1}-1} \right),$$

where we set $\tau_{|\boldsymbol{\tau}|+1} \equiv H + 1$. To proceed from here, we first observe that for any $m \in [M]$ and $i \geq 0$ such that $p_m^{1,i} > 0$,

$$\pi_{\tau_i+1:\tau_{i+1}} \cdot l^i_{\tau_i+1:\tau_{i+1}-1} \leq \pi_{\tau_i+1:\tau_{i+1}} \frac{\prod_{t=\tau_i+1}^{\tau_{i+1}-1} P_m(y_t|x_t)}{P_m^{\nu(\psi_i;\tau_i)}(s_{\tau_{i+1}}|s_{\tau_i+1})}$$

$$\leq \pi(a_{\tau_{i+1}}|h_{\tau_{i+1}}) \cdot \frac{\prod_{t=\tau_i+1}^{\tau_{i+1}-1} \pi(a_t|h_t) P_m(y_t|x_t)}{P_m^{\nu(\psi_i;\tau_i)}(s_{\tau_{i+1}}|s_{\tau_i+1})}$$

$$= \frac{P_m^{\pi}(x_{\tau_i+1:\tau_{i+1}}, y_{\tau_i+1:\tau_{i+1}-1}|h_{\tau_i+1})}{P_m^{\nu(\psi_i;\tau_i)}(s_{\tau_{i+1}}|s_{\tau_i+1})}. \qquad (21)$$

Thus, we can summarize that

$$\sum_{x_{1:H}} \sum_{y_{1:H}} \pi_{1:H} \left| \sum_{m=1}^M w_m^1 \prod_{t=0}^H P_m^1(y_t|x_t) - \sum_{m=1}^{M_2} w_m^2 \prod_{t=0}^H P_m^2(y_t|x_t) \right|$$

$$\leq \sum_{\boldsymbol{\tau} \in \mathtt{Subseq}(H,d)} \sum_{x_{\boldsymbol{\tau}}, y_{\boldsymbol{\tau}}} \Delta(x_{\boldsymbol{\tau}}, y_{\boldsymbol{\tau}}) \times \sum_{\substack{x_{0:\tau_1-1} \\ y_{0:\tau_1-1}}} \cdots \sum_{\substack{x_{\tau_{|\boldsymbol{\tau}|}+1:H} \\ y_{\tau_{|\boldsymbol{\tau}|}+1:H}}} \prod_{i=0}^{|\boldsymbol{\tau}|} \left( \frac{P_{m(x_{\boldsymbol{\tau}}, y_{\boldsymbol{\tau}})}^\pi (x_{\tau_i+1:\tau_{i+1}}, y_{\tau_i+1:\tau_{i+1}-1}|h_{\tau_i+1})}{P_{m(x_{\boldsymbol{\tau}}, y_{\boldsymbol{\tau}})}^{\nu(\psi_i;\tau_i)}(s_{\tau_{i+1}}|s_{\tau_i+1}, \phi)} \right).$$

We note that each term in the product sequence is equivalent to

$$\frac{P_{m(x_{\boldsymbol{\tau}}, y_{\boldsymbol{\tau}})}^\pi (h_{\tau_{i+1}+1}|h_{\tau_i+1})}{P_{m(x_{\boldsymbol{\tau}}, y_{\boldsymbol{\tau}})}^{\nu(\psi_i;\tau_i)}(s_{\tau_{i+1}}|s_{\tau_i+1}) P_{m(x_{\boldsymbol{\tau}}, y_{\boldsymbol{\tau}})}(y_{\tau_{i+1}}|x_{\tau_{i+1}})},$$

and thus

$$\prod_{i=0}^{|\boldsymbol{\tau}|} \left( \frac{P_{m(x_{\boldsymbol{\tau}}, y_{\boldsymbol{\tau}})}^\pi (x_{\tau_i+1:\tau_{i+1}}, y_{\tau_i+1:\tau_{i+1}-1}|h_{\tau_i+1})}{P_{m(x_{\boldsymbol{\tau}}, y_{\boldsymbol{\tau}})}^{\nu(\psi_i;\tau_i)}(s_{\tau_{i+1}}|s_{\tau_i+1})} \right)$$

$$= \left( \frac{P_{m(x_{\boldsymbol{\tau}}, y_{\boldsymbol{\tau}})}^\pi (x_{1:H}, y_{1:H})}{\prod_{i=0}^q P_{m(x_{\boldsymbol{\tau}}, y_{\boldsymbol{\tau}})}^{\nu(\psi_i;\tau_i)}(s_{\tau_{i+1}}|s_{\tau_i+1}) P_{m(x_{\boldsymbol{\tau}}, y_{\boldsymbol{\tau}})}(y_{\tau_{i+1}}|x_{\tau_{i+1}})} \right).$$

Here we set $\tau_{q+1} = H + 1$ and $P_m^\pi(s_{H+1}|\cdot) = 1$ for any $m, \pi$ and conditional event. Since the denominator does not depend on events outside of event-logging time-steps $\boldsymbol{\tau}$, we can marginalize the probabilities in the inner summation and conclude the lemma. $\square$

## E.4 Proof of Lemma A.3

Let us slightly extend the lemma such that we consider different sets of behavioral policies for different checkpoint time-steps.

*Proof.* Note that for all $m, s', s, t_1, t_2$, by the construction of $\psi_\xi$, it follows that

$$P_m^{\nu(\psi_\xi;t_1)}(s_{t_2} = s|s_{t_1}' = s') \geq \max_{\psi \in \Psi_\xi} \frac{P_m^{\nu(\psi;t_1)}(s_{t_2} = s|s_{t_1}' = s')}{n}.$$

We can observe that for any $m$,

$$P_m^{\nu(\boldsymbol{\psi}_\xi;\boldsymbol{\tau}, z(I;\boldsymbol{\tau}))}(s_{\boldsymbol{\tau}_I}', x_{\boldsymbol{\tau}/I}, y_{\boldsymbol{\tau}/I}) = \prod_{i \in I} P_m^{\nu(\psi_\xi;\tau_{i-1})}(s_{\tau_i+1}|s_{\tau_{i-1}}') \cdot \prod_{i \in [q]/I} \frac{1}{A} \cdot P_m^{\nu(\psi_\xi;\tau_{i-1})}(s_{\tau_i}|s_{\tau_{i-1}}') P_m(y_{\tau_i}|x_{\tau_i}).$$

On the other hand, for any $\pi \in \Pi$, $\boldsymbol{\tau} \subseteq [H]$, $I \in [|\boldsymbol{\tau}|]$, $x_{\boldsymbol{\tau}}, y_{\boldsymbol{\tau}}$,

$$P_m^\pi(s_{\boldsymbol{\tau}_I}', x_{\boldsymbol{\tau}/I}, y_{\boldsymbol{\tau}/I}) \leq \prod_{i \in I} \max_{\mathcal{T}_{1:\tau_{i-1}}} P_m^\pi(s_{\tau_i+1}|s_{\tau_{i-1}}', \mathcal{T}_{1:\tau_{i-1}}) \cdot \prod_{i \in [q]/I} \max_{\mathcal{T}_{1:\tau_{i-1}}} P_m^\pi(x_{\tau_i}|s_{\tau_{i-1}}', \mathcal{T}_{1:\tau_{i-1}}) P_m(y_{\tau_i}|x_{\tau_i}).$$

Applying the inequality with the definition of $\rho(\Psi_\xi; \pi)$, we have

$$C(\boldsymbol{\psi}_\xi; \pi) = \max_{\boldsymbol{\tau} \subseteq [H], |\boldsymbol{\tau}| \leq d} \max_{I \subseteq [|\boldsymbol{\tau}|]} \max_{\substack{\boldsymbol{s}' \in \mathcal{S}^{\otimes|I|} \\ (\boldsymbol{x}, \boldsymbol{y}) \in (\mathcal{X}, \mathcal{Y})^{\otimes|\boldsymbol{\tau}|-|I|}}} \max_{m \in [M]} \frac{P_m^\pi(s_{\boldsymbol{\tau}_I}' = \boldsymbol{s}', x_{\boldsymbol{\tau}/I} = \boldsymbol{x}, y_{\boldsymbol{\tau}/I} = \boldsymbol{y})}{P_m^{\nu(\boldsymbol{\psi}_\xi;\boldsymbol{\tau}, z(I;\boldsymbol{\tau}))}(s_{\boldsymbol{\tau}_I}' = \boldsymbol{s}', x_{\boldsymbol{\tau}/I} = \boldsymbol{x}, y_{\boldsymbol{\tau}/I} = \boldsymbol{y})}$$

$$\leq \max_{\boldsymbol{\tau} \subseteq [H], |\boldsymbol{\tau}| \leq d} \max_{I \subseteq [|\boldsymbol{\tau}|]} \max_{\substack{\boldsymbol{s}' \in \mathcal{S}^{\otimes|I|} \\ (\boldsymbol{x}, \boldsymbol{y}) \in (\mathcal{X}, \mathcal{Y})^{\otimes|\boldsymbol{\tau}|-|I|}}} \max_{m \in [M]} \prod_{i \in I} \frac{\max_{\mathcal{T}_{1:\tau_{i-1}}} P_m^\pi(s_{\tau_i+1}|s_{\tau_{i-1}}', \mathcal{T}_{1:\tau_{i-1}})}{P_m^{\nu(\psi_\xi;\tau_{i-1})}(s_{\tau_i+1}|s_{\tau_{i-1}}')}$$

$$\times \prod_{i \in [q]/I} A \cdot \frac{\max_{\mathcal{T}_{1:\tau_{i-1}}} P_m^\psi(s_{\tau_i}|s_{\tau_{i-1}}', \mathcal{T}_{1:\tau_{i-1}})}{P_m^{\nu(\psi_\xi;\tau_{i-1})}(s_{\tau_i}|s_{\tau_{i-1}}')}$$

$$\leq (nA \cdot \rho(\Psi_\xi; \pi))^d,$$

concluding Lemma A.3. $\square$

### E.5   Proof of Lemma A.4

This is by now a standard MLE technique for constructing confidence sets in RL [1].

*Proof.*   The proof follows a Chernoff bound type of technique:

$$\mathbb{P}_{\theta^*}\left(\sum_{(\tau,\pi)\in\mathcal{D}^k}\log\left(\frac{\mathbb{P}_\theta^\pi(\tau)}{\mathbb{P}_{\theta^*}^\pi(\tau)}\right)\geq\mathbb{E}_{\theta^*}\left[\sum_{(\tau,\pi)\in\mathcal{D}^k}\log\left(\frac{\mathbb{P}_\theta^\pi(\tau)}{\mathbb{P}_{\theta^*}^\pi(\tau)}\right)\right]+\beta\right)$$

$$\leq\mathbb{P}_{\theta^*}\left(\exp\left(\sum_{(\tau,\pi)\in\mathcal{D}^k}\log\left(\frac{\mathbb{P}_\theta^\pi(\tau)}{\mathbb{P}_{\theta^*}^\pi(\tau)}\right)\right)\geq\exp\left(\beta\right)\right)$$

$$\leq\mathbb{E}_{\theta^*}\left[\exp\left(\sum_{(\tau,\pi)\in\mathcal{D}^k}\log\left(\frac{\mathbb{P}_\theta^\pi(\tau)}{\mathbb{P}_{\theta^*}^\pi(\tau)}\right)\right)\right]\exp(-\beta).$$

The last inequality is by Markov's inequality. Note that random variables are $(\tau,\pi)$ in the trajectory dataset $\mathcal{D}$, and

$$\mathbb{E}_{\theta^*}\left[\sum_{(\tau,\pi)\in\mathcal{D}^k}\log\left(\frac{\mathbb{P}_\theta^\pi(\tau)}{\mathbb{P}_{\theta^*}^\pi(\tau)}\right)\right]=-\mathrm{KL}(\mathbb{P}_{\theta^*}(\mathcal{D}^k)||\mathbb{P}_\theta(\mathcal{D}^k))\leq 0.$$

Therefore,

$$\mathbb{E}_{\theta^*}\left[\exp\left(\sum_{(\tau,\pi)\in\mathcal{D}^k}\log\left(\frac{\mathbb{P}_\theta^\pi(\tau)}{\mathbb{P}_{\theta^*}^\pi(\tau)}\right)\right)\right]=\mathbb{E}_{\theta^*}\left[\Pi_{(\tau,\pi)\in\mathcal{D}^k}\frac{\mathbb{P}_\theta^\pi(\tau)}{\mathbb{P}_{\theta^*}^\pi(\tau)}\right]=\sum_{\mathcal{D}^k}\mathbb{P}_\theta(\mathcal{D}^k)=1.$$

Combining the above, taking a union bound over $k\in[K]$ rounds and $\theta\in\Theta$, letting $\beta=\log(K|\Theta|/\eta)$, with probability $1-\eta$, the inequality in Lemma A.4 holds. $\qquad\square$

### E.6   Proof of Lemma A.5

*Proof.*   By the TV-distance and Hellinger distance relation, for any $\iota,\tau,\pi$ and $t\in[H]$,

$$\mathrm{TV}^2\left(\mathbb{P}_\theta^\pi(\tau),\mathbb{P}_{\theta^*}^\pi(\tau)\right)\leq 2\mathrm{H}^2\left(\mathbb{P}_\theta^\pi(\tau),\mathbb{P}_{\theta^*}^\pi(\tau)\right)$$

$$=2\left(1-\mathbb{E}_{\tau\sim\mathbb{P}_{\theta^*}^\pi}\left[\sqrt{\frac{\mathbb{P}_\theta^\pi(\tau)}{\mathbb{P}_{\theta^*}^\pi(\tau)}}\right]\right)\leq-2\log\left(\mathbb{E}_{\tau\sim\mathbb{P}_{\theta^*}^\pi}\left[\sqrt{\frac{\mathbb{P}_\theta^\pi(\tau)}{\mathbb{P}_{\theta^*}^\pi(\tau)}}\right]\right).$$

To bound the summation over samples, we start from

$$\sum_{(\tau,\pi)\in\mathcal{D}^k}\mathrm{H}^2\left(\mathbb{P}_\theta^\pi(\tau),\mathbb{P}_{\theta^*}^\pi(\tau)\right)\leq-\sum_{(\tau,\pi)\in\mathcal{D}^k}\log\left(\mathbb{E}_{\tau\sim\mathbb{P}_{\theta^*}^\pi}\left[\sqrt{\frac{\mathbb{P}_\theta^\pi(\tau)}{\mathbb{P}_{\theta^*}^\pi(\tau)}}\right]\right).$$

On the other hand, by the Chernoff bound,

$$\mathbb{P}_{\theta^*}\left(\sum_{(\tau,\pi)\in\mathcal{D}^k}\log\left(\sqrt{\frac{\mathbb{P}_\theta^\pi(\tau)}{\mathbb{P}_{\theta^*}^\pi(\tau)}}\right)\geq\sum_{(\tau,\pi)\in\mathcal{D}^k}\log\mathbb{E}_{\tau\sim\mathbb{P}_{\theta^*}^\pi}\left[\sqrt{\frac{\mathbb{P}_\theta^\pi(\tau)}{\mathbb{P}_{\theta^*}^\pi(\tau)}}\right]+\beta\right)$$

$$\leq\mathbb{E}_{\theta^*}\left[\frac{\exp\left(\sum_{(\tau,\pi)\in\mathcal{D}^k}\log\left(\sqrt{\frac{\mathbb{P}_\theta^\pi(\tau)}{\mathbb{P}_{\theta^*}^\pi(\tau)}}\right)\right)}{\exp\left(\sum_{(\tau,\pi)\in\mathcal{D}^k}\log\mathbb{E}_{\tau\sim\mathbb{P}_{\theta^*}^\pi}\left[\sqrt{\frac{\mathbb{P}_\theta^\pi(\tau)}{\mathbb{P}_{\theta^*}^\pi(\tau)}}\right]\right)}\right]\exp(-\beta)$$

$$=\mathbb{E}_{\theta^*}\left[\frac{\Pi_{(\tau,\pi)\in\mathcal{D}^k}\sqrt{\frac{\mathbb{P}_\theta^\pi(\tau)}{\mathbb{P}_{\theta^*}^\pi(\tau)}}}{\Pi_{(\tau,\pi)\in\mathcal{D}^k}\mathbb{E}_{\tau\sim\mathbb{P}_{\theta^*}^\pi}\left[\sqrt{\frac{\mathbb{P}_\theta^\pi(\tau)}{\mathbb{P}_{\theta^*}^\pi(\tau)}}\right]}\right]\exp(-\beta)$$

$$= \mathbb{E}_{\theta^*} \left[ \mathbb{E}_{\theta^*} \left[ \frac{\Pi_{(\tau,\pi)\in\mathcal{D}^{k-1}} \sqrt{\frac{\mathbb{P}_\theta^\pi(\tau)}{\mathbb{P}_{\theta^*}^\pi(\tau)}}}{\Pi_{(\tau,\pi)\in\mathcal{D}^{k-1}} \mathbb{E}_{\tau\sim\mathbb{P}_{\theta^*}^\pi} \left[ \sqrt{\frac{\mathbb{P}_\theta^\pi(\iota,\tau)}{\mathbb{P}_{\theta^*}^\pi(\iota,\tau)}} \right]} \middle| \mathcal{D}^{k-1} \right] \right] \exp(-\beta)$$

$$= \mathbb{E}_{\theta^*} \left[ \frac{\Pi_{(\tau,\pi)\in\mathcal{D}^{k-1}} \sqrt{\frac{\mathbb{P}_\theta^\pi(\tau)}{\mathbb{P}_{\theta^*}^\pi(\tau)}}}{\Pi_{(\tau,\pi)\in\mathcal{D}^{k-1}} \mathbb{E}_{\tau\sim\mathbb{P}_{\theta^*}^\pi} \left[ \sqrt{\frac{\mathbb{P}_\theta^\pi(\iota,\tau)}{\mathbb{P}_{\theta^*}^\pi(\iota,\tau)}} \right]} \right] \exp(-\beta) = ... = \exp(-\beta),$$

where in the last line, we used the tower property of expectation. Thus, again by setting $\beta = \log(K|\Theta|/\eta)$, with probability at least $1 - \eta$, we have

$$\sum_{(\tau,\pi)\in\mathcal{D}^k} \mathtt{H}^2(\mathbb{P}_\theta^\pi(\tau), \mathbb{P}_{\theta^*}^\pi(\tau)) \leq -\frac{1}{2} \sum_{(\tau,\pi)\in\mathcal{D}^k} \log\left( \frac{\mathbb{P}_\theta^\pi(\tau)}{\mathbb{P}_{\theta^*}^\pi(\tau)} \right) + \beta$$

$$= -\frac{1}{2} \sum_{(\tau,\pi)\in\mathcal{D}^k} \log\left( \frac{\mathbb{P}_\theta^\pi(\tau)}{\mathbb{P}_{\theta^*}^\pi(\tau)} \right) + \frac{1}{2} \sum_{(\tau,\pi)\in\mathcal{D}^k} \log\left( \frac{\mathbb{P}_\theta^\pi(\tau)}{\mathbb{P}_{\theta^*}^\pi(\tau)} \right) + \beta,$$

for all $k \in [K]$ and $\theta \in \Theta$. Now we can apply Lemma A.4, and finally have

$$\sum_{(\tau,\pi)\in\mathcal{D}^k} \mathtt{H}^2(\mathbb{P}_\theta^\pi(\tau), \mathbb{P}_{\theta^*}^\pi(\tau)) \leq -\frac{1}{2} \sum_{(\tau,\pi)\in\mathcal{D}^k} \log\left( \frac{\mathbb{P}_\theta^\pi(\tau)}{\mathbb{P}_{\theta^*}^\pi(\tau)} \right) + \frac{3}{2}\beta.$$

Since $\mathtt{TV}^2 \leq 2\mathtt{H}^2$, we get the lemma. $\qquad\square$

