# OpenReview forum: "RL in Latent MDPs is Tractable: Online Guarantees via Off-Policy Evaluation"
_NeurIPS.cc/2024/Conference — NeurIPS 2024 poster_

### Official Review · Reviewer_Ch5D · 2024-07-09

**Soundness:** 3
**Presentation:** 3
**Contribution:** 3
**Rating:** 6
**Confidence:** 3

**Summary:**

This paper studies latent Markov decision processes (LMDP) with M = O(1) number of MDPs. In other words, there exists a set of MDPs (unknown to the agent) and the environment randomly selects one MDP at the beginning of each episode. The selected MDP is not revealed to the agent and therefore the agent must infer which MDP is selected from the feedback. This paper proposes an algorithm with sample complexity poly(S,A)^M that matches the \Omega((SA)^M) lower bound.

On the technical side, this paper designs a model-based exploration algorithm that actively collects new data if the collected data can shrink the confidence interval of the model parameters. On standard tabular RL, this paper shows that (a) the algorithm terminates after poly(SAH) rounds of exploration, and (b) any remaining model parameter can be used to construct a near-optimal policy after the exploration stage. Because of the generality of the algorithm, this paper further extends the algorithm to the LMDP setting and proves a poly(S,A)^M sample complexity.

**Strengths:**

-	The exposition of the main ideas is very clear because of the instantiation on tabular MDPs.
-	The OMLE algorithm is neat, and it provides a general framework for model-based exploration.
-	The sample complexity result matches the \Omega(S,A)^M lower bound and requires no additional assumption on the structure of the LMDP family.

**Weaknesses:**

-	The algorithm is not computationally efficient even for standard tabular MDPs because it has to enumerate both the parameters and the policy in the confidence interval, though I understand that computational efficiency is not the focus of this paper.
-	Section 4 is hard to follow without background knowledge of several prior works. E.g., the choice d = 2M-1 is not well-justified. I encourage the authors to revise Section 4 to make it more self-contained.
-	It seems to me that the title of this paper somewhat overclaims the result since the sample complexity is still exponential in M, and the assumption that M is finite is not intrinsic to the LMDP setup (although it is necessary due to the sample complexity lower bound).

**Questions:**

-	It seems to me that the OMLE algorithm is very general. Is it straightforward to prove some general sample complexity results with problem-dependent bounds? Intuitively, the bound would include terms like the statistical complexity of the model class and some complexity measure of the MDP family.
-	The data collection step in Algorithm 2 (Line 5-7) seems to be quite brute-force since it has to enumerate all the segmented policy. Could the authors elaborate on its reason?

**Limitations:**

The authors adequately addressed the limitations and potential negative societal impact.

---

> ### Author Rebuttal · Authors · 2024-08-06
>
> We thank the reviewer for the thoughtful review and constructive feedback on our paper. We address the mentioned weaknesses below.
>
>
> **Practicality of the Proposed Algorithm**
>
> As the reviewer correctly pointed out, our focus was indeed on the purely statistical learning aspect of the problem. Nevertheless, we acknowledge and appreciate the reviewer's perspective on the computational aspects of the algorithm. Our long-term goal is to design an oracle-efficient algorithm for LMDPs that can be implemented with general function classes and ERM-style oracles. While we believe this is a feasible direction, achieving it will require significant effort from both theoretical and practical standpoints.
>
>
> **Intuition on Policy-Segmentation Regarding Prior Work**
>
> Thank you for your suggestion. Indeed, we agree that it would be useful to provide the intuition on how we connect moment-exploration ideas presented in [1] to our off-policy evaluation results. We will add the following paragraph in our revision at the beginning of Section 4:
>
> *Before we dive into our key results, let us provide our intuition on how we construct the OPE lemma for LMDPs. Our construction is inspired by the moment-exploration algorithm proposed in [1]: when state-transition dynamics are identical across latent contexts, {\it i.e.,} $T_1 = T_2 = ... = T_M$, we can first learn the transition dynamics with any reward-free type exploration scheme for MDPs [2], and then set the exploration policy that sufficiently visits certain tuples of state-actions of length at most $d=2M-1$. For the exploration policy, the work in [1] set a memoryless exploration policy $\psi \in \Pi_{\texttt{mls}}$ which sets the visitation probability to certain tuples sufficiently large. We note that the same moment-exploration strategy cannot be applied to general LMDPs with different state-transition dynamics since learning the transition dynamics itself involves latent contexts. Nevertheless, the intuition from [1] suggests that our key statistics are this visitation probabilities to all tuples of state-actions within a trajectory.*
>
>
>
> **Significance of Our Results on $M$**
>
> We accept this criticism and appreciate the feedback. That being said, we would like to highlight that our results represent a substantial advancement from previous work on LMDPs. Specifically, our algorithm is the first to achieve sample efficiency in a partially observed setup without relying on standard assumptions such as weakly-revealing or low-rankness, which are well-studied in the literature. Even in cases where $M=2$ or $3$, the solutions to LMDPs were previously unknown, making our findings significant. While we acknowledge that our current results are achieved with $M=O(1)$, we believe it is important to emphasize the positive aspect of our contributions in pushing the boundaries of what is known in this field.
>
>
>
>
>
>
> **Possible Extensions to General Function Classes**
>
> This is a great question. We assume that by "problem-dependent bound", the reviewer is referring to sample-complexity bounds with general function approximation. We believe that extending our approach to general function approximation is feasible, particularly given recent advances in RL; however, this is not entirely straightforward and requires careful consideration. For Latent MDPs specifically, our starting point will be to define what is the good notion of "coverage" in LMDPs with large state/action spaces -- that will direct us to the question of what are the "moments" with general function approximation. If we can define this notion well, then we believe that we should be able to come up with a notion similar to the LMDP-coverage with function approximation. Another possibility is to come up with a general notion of statistical complexity measure for Latent MDPs/POMDPs such as DEC [3] (see also our discussion on GEC with Reviewer 8u1Y). Such an extension is beyond the scope of this paper, and we think it is an exciting future direction.
>
>
>
>
>
>
> **More Efficient Ways to Construct Data-Collection Policies**
>
> This is another great point. The current brute-force construction is due to the nature of worst-case analysis, where joint events at all length-$d$ tuples of state-actions need to be observed under all contexts. Since we do not have the observability over which policy drives us to the next state-action pair under latent contexts of interest, our solution brute-force all possibilities.
>
> That being said, we are hopeful that under some practical assumptions, we may expect improvement in two ways:
>
> - We can rule out some redundant combinations, as they might already be covered by other combination (e.g., this idea can be related to G-optimal design as in the Latent Multi-Armed Bandit setting [4])
>
> - Other possibility is to identify that for some combinations, at some point, it becomes obvious that the segmentation of significantly smaller than $d$ segments would be sufficient.
>
> These are the next big questions and important future work.
>
>
>
> ------------------------
>
>
> We hope that our answers to the raised questions are satisfactory. Please let us know if you have any other concerns or questions. We would greatly appreciate it if you could consider reevaluating our work, taking into account the strengths and improvements we've outlined.
>
>
> [1] Kwon et al., "Reward-mixing MDPs with few latent contexts are learnable", ICML 2023
>
> [2] Jin et al., "Reward-free exploration for reinforcement learning", ICML 2020
>
> [3] Foster et al., "The statistical complexity of interactive decision making", arXiv 2021
>
> [4] Kwon et al., "Tractable Optimality in Episodic Latent MABs", NeurIPS 2022

---

> > ### Comment · Reviewer_Ch5D · 2024-08-13
> >
> > Thank you for the response. I will keep my original evaluation.

---

> > > ### Author Response · Authors · 2024-08-13
> > > **Thank you for the response**
> > >
> > > Thank you for the feedback!

---

### Official Review · Reviewer_AN8Q · 2024-07-10

**Soundness:** 4
**Presentation:** 3
**Contribution:** 4
**Rating:** 8
**Confidence:** 3

**Summary:**

This paper studies latent MDP, where the underlying dynamics and rewards are controlled by some latent states (not revealed to the learner), and the learner attempts to do learning and planning based on the trajectory data.

The algorithm builds from the optimistic MLE algorithm, which iteratively checked whether whether there exists uncovered policies. To make the OMLE works for LMDP, this paper constructed segmented policies, and iteratively check whether such policies are all covered.

As a result, they showed the first polynomial sample complexity result for LMDP when the number of latent states are constant.

**Strengths:**

The paper is well written. The algorithm, theorems and proofs are clear.

The results of polynomial sample complexity for LMDP with constant latent states seem very interesting.

The idea of constructing segmented policies is novel.

**Weaknesses:**

I don't see any significant weaknesses of this paper.

**Questions:**

Can these results somehow adapt to the POMDP setting?

**Limitations:**

Yes. The authors addressed all the limitations listed in the guidelines.

---

> ### Author Rebuttal · Authors · 2024-08-06
>
> We sincerely appreciate the encouraging comments and positive assessment of our paper. Below, we address an interesting question you raised:
>
> **Can these results somehow adapt to the POMDP setting?**
>
> Yes, we believe our results can be adaptable to the POMDP setting in two senses:
>
> - Algorithmically speaking, our algorithm overall falls in a general framework of OMLE; the key difference is in the data-collection policy part which may differ with different environments and assumptions. But at a higher-level, they can be viewed in a unified principle.
>
> - Technically speaking, our analysis based on the notion of coverage can be useful to give a different interpretation of solving the POMDP settings (that are known to be tractable). The key to understand in such a way is to define a suitable notion of coverage in general POMDPs -- which is in part done in some recent work on off-policy learning for POMDP [1]. This is an interesting future question.
>
>
>
>
> --------------------
>
> Thank you once again for your thoughtful review and positive assessment. If you have any further questions or suggestions, please let us know.
>
> [1] Zhang and Jiang, "On the curses of future and history in future-dependent value functions for off-policy evaluation", arXiv 2024

---

> > ### Comment · Reviewer_AN8Q · 2024-08-12
> >
> > Thank you for your response. I don't have further questions.

---

> > > ### Author Response · Authors · 2024-08-13
> > > **Thank you for the response**
> > >
> > > Thanks for the response and positive feedback!

---

### Official Review · Reviewer_8u1Y · 2024-07-16

**Soundness:** 3
**Presentation:** 2
**Contribution:** 1
**Rating:** 3
**Confidence:** 3

**Summary:**

This paper introduces a new version of the coverage coefficient for analyzing latent Markov Decision Processes (MDPs). It demonstrates how to link the proposed coverage coefficient with sample complexity using MDPs. Additionally, the paper presents an algorithm and provides a bound on the sample complexity of this algorithm.

**Strengths:**

The use of MDPs to illustrate the concept is effective, making it easier for the audience to follow.

**Weaknesses:**

1. The comparison with related work is insufficient. Including a table that compares the results in this paper with those in [1] would be helpful.

2. The connection and comparison between this work and existing studies on the coverage coefficient [2] should be more concrete.

3. The meaning and design of the segment policy are unclear, making it difficult for the audience to understand the intuition behind why this coverage coefficient is helpful for analysis. Although a counterexample is provided in the appendix, it is still not intuitive enough.

4. In my understanding, the core idea behind the analysis is that complexity depends on the longest policy sequence $\pi_1,\ldots,\pi_k$, where $C(\pi_i,\pi_j)=\infty$. If this understanding is correct, there are two follow-up questions:
(a) Can we define a complexity measure similar to the Eluder dimension?
(b) Can the complexity measure generalize to cases beyond latent MDPs?
(c) Is this complexity measure equivalent to or weaker than the Eluder dimension?
Answers to these questions might help improve this paper.

5. Although the results in this paper do not require further structural assumptions, the significance of an exponential upper bound is questionable.

6. Lack of discussion with works that have complexity measure without Markovian assumption [3].

[1]. J. Kwon, Y. Efroni, C. Caramanis, and S. Mannor. Reinforcement learning in reward-mixing MDPs. Advances in Neural Information Processing Systems, 34, 2021.

[2]. P. Amortila, D. J. Foster, and A. Krishnamurthy. Scalable online exploration via coverability. arXiv preprint arXiv:2403.06571, 2024.

[3]. Zhong, Han, et al. "Gec: A unified framework for interactive decision making in mdp, pomdp, and beyond."

**Questions:**

See "Weaknesses" part for the questions.

**Limitations:**

See "Weaknesses" part for the limitations.

---

> ### Author Rebuttal · Authors · 2024-08-06
>
> We are grateful for your insights and suggestions, which will help us improve our work. Below, we address the weaknesses you mentioned.
>
>
> **Comparison to Previous Work on LMDPs**
>
> We would like to highlight that our work is the first to propose a general exploration algorithm applicable to the entire class of LMDP models. In contrast, previous works [1] focus solely on the subclass of LMDPs known as "Reward-Mixing" MDPs. These models assume no context ambiguity in transition dynamics, only in reward models. Under this assumption, data collection becomes less challenging. One can first learn the transition model separately through reward-free exploration, and then explore specific state-action tuples in order to obtain samples of "moments". However, when transition dynamics also vary across contexts, data-collection becomes significantly more complex, and is the main challenge in our work.
>
> We will further clarify this distinction and the associated challenges in Section 1.1 - Challenge 2 in our revised manuscript.
>
>
>
> **Coverage Coefficient in MDPs vs LMDPs**
>
> Existing work that studies "the role of coverage in online exploration" is focused on fully observed settings (Low-Rank MDP, Block MDP are also fully-observed, although latent dynamics is much lower dimensional). In contrast, LMDP, and more broadly POMDP, involve dynamics and policies that are history-dependent -- an optimal policy of an LMDP may depend on the entire history. This means that a single latent-state coverability does not adequately capture the complexity of offline learning in these settings (as we illustrated in the counterexample in Appendix).
>
> Consequently, off-policy evaluation in LMDPs requires a new notion of coverage that measures coverability over *sequences of state-action pairs*. Our main contribution is the proposal of this LMDP-coverage concept, which we connect to off-policy evaluation and online exploration. We believe this not only represents a significant advancement in learning LMDPs but also offers a conceptual contribution to the learning of general POMDPs. We will make this point clearer and more accessible to readers in our revision.
>
>
>
>
>
>
> **Eluder Dimension vs Our Approach**
>
> The Eluder dimension essentially measures the longest possible policy sequence that can incur a large prediction error. However, in partially observed settings, this definition may fail to capture the true complexity of online exploration. For instance, in multi-step weakly-revealing POMDPs (or PSRs), without executing core-tests, we may still suffer from the curse of horizon [2].
>
> One might modify the definition of the Eluder dimension as proposed in the GEC paper you shared (details will follow below). However, similarly to the limitation of known tractable POMDP classes, the GEC definition also relies on the prior knowledge of how to construct the data-collection policies (using core-tests). Our technical contribution lies in the design and understanding of data-collection policies in LMDPs. We believe there is significant potential to develop a more general, fine-grained notion of complexity for a broader class, potentially including both revealing POMDPs and LMDPs. We see this as an exciting direction for future research.
>
>
>
>
>
>
> **Upper Bounds Exponential in M?**
>
> We accept this criticism, though due to the lower bound established in [3] we cannot avoid this dependence in M without any assumptions. However, we would like to highlight that our results represent a substantial advancement from previous work. Specifically, our algorithm is the first to achieve sample efficiency in a partially observed setup without relying on standard assumptions such as weakly-revealing or low-rankness, which are well-studied in the literature. Even in cases where $M=2$ or $3$, the solutions to LMDPs were previously unknown, making our findings significant. While we acknowledge that our current results are achieved with $M=O(1)$, we believe it is important to emphasize the positive aspect of our contributions in pushing the boundaries of what is known in this field (we also refer the reviewer to our general response on the importance of studying LMDPs).
>
>
>
>
>
> **Comparison/Connection to Other Complexity Measures (GEC)**
>
> We thank the reviewer for drawing our attention to this work, and we will add this reference in our revision. Yes, the idea behind GEC resonates with us at a very high-level -- it aims to capture the largest discrepancy in prediction errors from small training errors. However, as with other complexity measures such as (generalized) Eluder-dimension or Decision estimation coefficient, the main bottleneck is to show the boundedness of GEC in LMDPs, as there has been no upper bound established up-to-date. Furthermore, our construction of exploration (data-collection) policy is very different from known methods.
>
> While all the above is in part described in our Section 1.1, we will be more explicit about comparison to existing complexity measures in our revision.
>
>
> -------------
>
> We hope our responses clarify any misunderstandings and resolve the issues identified. Please let us know if there are any remaining questions or concerns. Otherwise, we kindly request a reevaluation of our work in light of the provided clarifications. Thank you for your consideration and effort.
>
>
>
> [1] Kwon et al., "Reinforcement Learning in Reward-Mixing MDPs", NeurIPS 2021
>
> [2] Chen et al., "Lower bounds for learning in revealing POMDPs", ICML 2023
>
> [3] Kwon et al., "RL for Latent MDPs: Regret Guarantees and a Lower Bound", NeurIPS 2021

---

### Official Review · Reviewer_jHm3 · 2024-07-26

**Soundness:** 3
**Presentation:** 3
**Contribution:** 2
**Rating:** 6
**Confidence:** 3

**Summary:**

The paper studies latent MDPs, an MDP framework with a set of MDPs and the environment samples a random MDP at the beginning of each episode. To avoid a $A^H$ sample complexity, previous works either assume separation or similarity of transitions. This work removes these conditions, and provide an algorithm with $(SA)^{O(M)}$ upper bound, which matches the lower bound with $M = O(1)$. Speficially, the algorithm adapts the information theoretic Optimistic MLE algorithm for the LMDP setting, by collecting data with all possible segments of all previous policies (+random actions). The key analysis tool is the LMDP coverage coefficient.

**Strengths:**

1. The paper proposes the first algorithm that matches the lower bound of LMDPs (to poly order) without the assumption of separation and similar transition dynamics.

2. The proposed LMDP coverage coefficient, although exponential in $d$ ($M$), seems to capture the right complexity of the LMDPs.

3. The detailed and intutive introduce of Optimistic MLE algorithm makes the proposed algorithm easy to understand.

**Weaknesses:**

1. The techincal innovation seems rather limited -- the main techiques are quite similar as OMLE.

2. I do not understand the relation to OPE: which part of the algorithm relates to OPE? It seems like the only change to OMLE is the way of data collection so there is no additional OPE subroutine?

3. The "without any additional structural assumption" claim in the abstract seems inaccurate: even OMLE is limited with the assumption of finite SAIL (bli-linear rank), and the proposed analysis only applies to tabular? Also, does the analysis generate to general function approximation if we ues the correponding coverage coefficient?

**Questions:**

see above

---

> ### Author Rebuttal · Authors · 2024-08-06
>
> We thank the reviewer for a thoughtful review and constructive feedback on our paper. We would like to start by emphasizing our technical novelty.
>
>
> **About Technical Novelties**
>
> Our LMDP-OMLE algorithm builds on the general framework of OMLE. The key novelty of the algorithm is the design of data-collection (exploration) policies (we also refer the reviewer to our general responses on the novelties). In designing the data-collection policy, we removed the need for core-tests and well-conditionedness. Further, in our analysis, we introduced the novel concept of LMDP coverage coefficient. These innovations result from efforts to formalize *moment-exploration*, as suggested in previous work [1]. Due to these, we were able to remove the restrictive assumptions of shared transition dynamics and separation.
>
>
> Hence, our key results deviate substantially from existing approaches, and, hopefully, establish a new perspective on exploration in partially observed environments. Further, we believe our results contain a sufficient amount of novel concepts and analysis techniques.
>
>
>
>
> **Off-Policy Evaluation for Online Guarantees**
>
> Our off-policy evaluation guarantee guides the data-collection policy of the LMDP-OMLE algorithm. In particular, the key novelty lies in the proposal of LMDP coverage coefficient (Definition 4.1): this quantity measures the data-coverage in terms of the visitation probability to *tuples* of different states and actions. Algorithmically, this suggests that our exploration algorithm should aim to visit "uncovered" tuples of state-action pairs, reaching one target state (and action) and moving to the next target state (and action) in the target tuple. Once we can design the exploration policy in such a way, we can show the sample-complexity upper bound via the coverage-doubling argument (Theorem 4.5) as we have illustrated in the MDP case.
>
>
> Here we note that establishing online guarantees via off-policy evaluation is just one possible approach for exploration in LMDPs. It would be very interesting to see if other approaches such as going through the problem-specific complexity measures such as Eluder-dimension [2], Bellman-rank [3] or Decision-Estimation Coefficient (DEC) [4] can be applied using the results provided in this work.
>
>
>
>
> **Why Our Approach Doesn't Make Any Assumptions?**
>
> We clarify that this work studies LMDP "without any assumptions" in the sense that we consider the class of *all possible LMDP models*, unlike in previous work that assumes certain separations [5,6] or shared transitions dynamics [1].
>
>
> -------------
>
> We hope our responses have clarified any doubts or questions. If there are any remaining issues, please let us know. Otherwise, we would highly appreciate it if the reviewer could consider raising the score based on our responses.
>
>
>
>
> [1] Kwon et al., "Reward-mixing MDPs with few latent contexts are
> learnable", ICML 2023
>
> [2] Russo and Van Roy, "Eluder dimension and the sample complexity of optimistic exploration", NeurIPS 2013
>
> [3] Jin et al., "Bellman eluder dimension: New rich classes of RL problems, and sample-efficient algorithms", NeurIPS 2021
>
> [4] Foster et al., "The statistical complexity of interactive decision
> making", arXiv 2021
>
> [5] Hallak et al., "Contextual markov decision processes", arXiv 2015
>
> [6] Chen et al., "Near-optimal learning and planning in separated
> latent MDPs", COLT 2024

---

> > ### Comment · Reviewer_jHm3 · 2024-08-13
> >
> > I appreciate the author's reponse. I still think the work is solid and my original evaluation is appropriate so I will maintain my original score. Also, it will also be helpful if the authors could revise the statement of "without any additional structural assumption" in the abstract.

---

> > > ### Author Response · Authors · 2024-08-13
> > > **Thank you for the response**
> > >
> > > Thank you for the feedback and the positive view on our contribution. Further, we will edit the abstract accordingly to clarify this point.

---

### Author Rebuttal · Authors · 2024-08-06

We thank all reviewers for their effort, time, and their valuable feedback. While we respond to each reviewer on the specific concerns raised, we would like to emphasize the importance of studying the Latent MDP setting as well as the technical novelties in our work.



**Why Solving LMDPs is Important:**


It is often useful to model the population of environments as a mixture of simpler distributions. Latent MDP is an interactive version of mixture modeling, and has a potential for many real world problems, *e.g.,* in dialogue, recommender or in healthcare systems, when complete information on a user or patient is not given. Nevertheless, no algorithm has been known beyond the setting when clustering is relatively straight-forward. We believe that studying general cases when clustering is not so straight-forward has many potentials to advancing the field both in theory and practice, as we detail below.




- *LMDP requires rethinking exploration strategies in partially observed environments:*  Several known approaches for general POMDPs overcome the challenge by assuming (1) the well-conditionedness of the system (latent state-observation emission matrix must be in full-rank), and (2) the prior knowledge of core-tests (those policies that guarantee the full rankness of the state-observation matrix). Not only does this require strong domain knowledge, but also there is no known approach to learn the optimal policy beyond such cases.



- *Potential to enlarge the scope beyond known tractable POMDP classes and general approaches:* Our work is the first to propose the sample-efficient algorithm for an important and broad class of POMDPs that do not provide the common assumptions of well-conditionedness and known core-tests. Furthermore, the proposed LMDP-OMLE algorithm is designed with as much flexibility and generality as possible without specific domain knowledge and assertions, unlike previous solutions that rely on certain clustering-enabling or separation assumptions. We believe our result paves the way to push existing theories further to classes of other potentially tractable POMDPs (in particular, to general function approximation with large state/action spaces in POMDPs).








**Technical Novelties in Our Work**

While we follow the principles of Optimistic Maximum Likelihood Estimation (OMLE) [1], we emphasize that our key novelty lies in the algorithmic design and analysis of data-collection policies given the confidence set, without the given knowledge of core-tests. Our approach is the first that goes beyond the known assumptions for POMDPs, and integrates the method-of-moments in the sequential decision-making setup. In particular, we formalize the concept of moment-matching in our off-policy evaluation lemma (Lemma 4.2) by introducing the visitation probability to tuples of different state-action pairs and learning their joint probabilities (this captures the correlations and moments of the system). Utilizing this off-policy evaluation lemma, we derive a sample-efficient algorithm through our coverage-doubling argument, detailed in Lemma D.1 of the Appendix. Although the mathematics involved is not particularly fancy and is purely algebraic, we believe that the conceptual and algorithmic innovations are substantial.


[1] Liu et al., "Optimistic MLE: A Generic Model-Based Algorithm for Partially Observable Sequential Decision Making", STOC 2023

---

### Decision · Program_Chairs · 2024-09-25

**Decision:**

Accept (poster)

**Comment:**

The paper presents a novel approach to latent Markov decision processes (LMDPs). The work's primary contributions include the introduction of the LMDP coverage coefficient and adapting the OMLE algorithm to this setting, offering a detailed analysis of sample complexity in tabular settings. Reviewers appreciated the novelty of the segmentation policies. However, some raised concerns about the limited technical innovation compared to existing methods like OMLE and a lack of sufficient comparison with related work. Despite these concerns, most reviewers lean towards acceptance.